# PyCHAM (v2.0.4): a Python box model for simulating aerosol chambers

Simon Patrick O'Meara[1,2], Shuxuan Xu[1], David Topping[1], M. Rami Alfarra[1,2], Gerard Capes[3], Douglas Lowe[3], Yunqi Shao[1], and Gordon McFiggans[1]

[1]Department for Earth and Environmental Sciences, University of Manchester, UK, M13 9PL
[2]National Centre for Atmospheric Science, University of Manchester, UK, M13 9PL
[3]Research Computing Services, University of Manchester, UK, M13 9PL

**Correspondence:** Gordon McFiggans (g.mcfiggans@manchester.ac.uk)

**Abstract.** In this paper the CHemistry with Aerosol Microphysics in Python (PyCHAM) box model software for aerosol chambers is described and assessed against benchmark simulations for accuracy. The model solves the coupled system of ordinary differential equations for gas-phase chemistry, gas-particle partitioning and gas-wall partitioning. Additionally, it can solve for coagulation, nucleation and particle loss to walls. PyCHAM is open source, whilst the graphical user interface, modular structure, manual, example plotting scripts and suite of tests for troubleshooting and tracking the effect of modifications to individual modules have been designed for optimal usability. In this paper, the modelled processes are individually assessed against benchmark simulations, and key parameters described. Examples of output when processes are coupled are also provided. Sensitivity of individual processes to relevant parameters is illustrated along with convergence of model output with increasing temporal resolution and number of size bins. The latter sensitivity analysis informs our recommendations for model setup. Where appropriate, parameterisations for specific processes have been chosen for their general applicability with their rationale detailed here. It is intended that PyCHAM aids the design and analysis of aerosol chamber experiments, with comparison of simulations against observations allowing improvement of process understanding that can be transferred to ambient atmosphere simulations.

## 1  Introduction

Many major advances in atmospheric modeling have arisen from chamber observations. For example, the partitioning of vapours to particles (Odum et al., 1996), the gas-phase chemistry of ozone as part of the Master Chemical Mechanism (MCM) (Jenkin et al., 1997), the gas-phase chemistries of limonene (Carslaw et al., 2012) and $\beta$-caryophyllene (Jenkin et al., 2012). Such advances can be incorporated into improved chamber models (e.g. Charan et al., 2019), aiding the design of experiments to interrogate further processes and systems (e.g. Peräkylä et al., 2020). As chamber use has multiplied, so too have chamber models, with many now published (Naumann, 2003; Pierce et al., 2008; Lowe et al., 2009; Roldin et al., 2014; Sunol et al.,

2018; Topping et al., 2018; Charan et al., 2019; Roldin et al., 2019). Chamber scientists without modelling expertise or access may be limited in the design, interpretation and advancement of both chamber experiments and their contribution to models. To address this requirement PyCHAM (CHemistry with Aerosol Microphysics in Python) has been developed in the framework of the EUROCHAMP2020 Simulation Chamber Research Infrastructure (Oliveri, 2018).

In this paper the processes represented in PyCHAM are described, along with details of software application. Where relevant, equations are presented and output from PyCHAM is compared against benchmark simulations to assess accuracy and determine whether calculations are performing as intended. It is not the intention of this paper to compare PyCHAM against observations, which is the focus of future work. In the following two sections the objectives, rationale and structure of the software are explained.

## 2  Purpose and scientific basis

Consistent with the criteria set by the EUROCHAMP2020 research project (Oliveri, 2018), PyCHAM is open source (available at https://github.com/simonom/PyCHAM), user-friendly and aims to be capable of representing the latest scientific understanding. It has been designed and tested on desktop computers for Windows, Linux and Mac operating systems. Python is the chosen language for two key reasons: code can be transferred between computers without the limitation of requiring a native or proprietary compiler (thereby improving ease of use and portability), and the relatively versatile parsing capability which allows the user to readily vary model inputs. The accessibility, usability and basic functionality of PyCHAM has been reviewed in O'Meara et al. (2020). The current paper presents a detailed description and introductory analysis of the PyCHAM functionality that was not the focus of O'Meara et al. (2020).

Aerosol chambers (interchangeably called smog chambers), defined as those used for interrogating gas- and particle-phase processes, provide a method for isolating specific processes of interest without the conflating effects present in the ambient atmosphere. Ultimately the goal of the chamber is to improve understanding and quantitative constraint on the evolution of the physicochemical properties of the gas- and particle-phase (Schwantes et al., 2017; Charan et al., 2019; Hidy, 2019). A description of chamber processes first requires consideration of the chamber characteristics, including: wall material (frequently fluorinated ethylene-propene film (FEP Teflon), though others are used), lighting, and dimensions. Two approaches are used to inlet components: batch mode whereby set volumes of gas or particle are injected at specific times, or in flow mode with a constant influx of gas or particle (Jaoui et al., 2014). The model variables input file for PyCHAM allows users to setup simulations for both modes along with other experiment descriptors that allow simulation of a broad range of chamber investigations: with or without seed particles; with or without nucleation; variable temperature, pressure and relative humidity; for illuminated experiments, either natural light intensity (for open roof chambers) or known actinic flux (for chambers with bulbs) that can be turned on and off at set times. The full introduction to model variables is given in Section 4.

Two previous models act as platforms on which PyCHAM developed: the Microphysical Aerosol Numerical model Incorporating Chemistry (MANIC) (Lowe et al., 2009) and PyBox (Topping et al., 2018), with the former guiding multi-phase processes and the latter guiding python parsing and automatic generation of chemical reaction modules. PyCHAM treats gas-

phase photochemistry, gas-particle and gas-wall partitioning, coagulation, nucleation and particle deposition to walls in zero dimensions. A key feature is its aim to be generally applicable, such that gas-wall partitioning, particle deposition to wall and nucleation - all processes with outstanding uncertainties - are parameterised and may be fitted to observations. Below we detail the constraint necessary for fitting the relevant parameters. The full list of PyCHAM applications is numerous and will increase as chamber experiments evolve. Key applications include designing chamber experiments, developing gas-phase photochem-

istry mechanisms, quantifying gas-wall partitioning parameters, developing nucleation models and interrogating the effects of processes on secondary particulate matter (SPM) evolution.

    The processes included in PyCHAM are typically a subset of those represented in large-scale (regional and global) atmospheric models and it is intended that once a process has been successfully modelled by PyCHAM it can be transferred, possibly via parameterisation, to a large-scale model for evaluation and application (as illustrated by the gas-particle partitioning and

gas-phase chemistry advances cited in the introduction). When interrogating the simulation of a given process it is necessary that conflating processes are modelled accurately, such that uncertainty around their effects is not compromising. Therefore, the main objective of this paper is to assess the accuracy and precision of the fundamental representation of the individual processes considered in PyCHAM through comparison with benchmark simulations. Comparison of some results presented below with experimental observations is possible, however it is beyond the scope of this paper to investigate the accuracy of chemical

mechanisms or estimation methods that PyCHAM can use, rather the examples below illustrate the utility of PyCHAM to test the sensitivity of such techniques.

    An example of PyCHAM application in which several major processes are influential is provided by simulating an experiment based on the role of nitrate radical ($NO_3$) oxidation of limonene in secondary organic particulate matter (SOPM) evolution (Fig. 1). Such an experiment has implications for indoor air quality at night time when the photolysis of $NO_3$ ceases (Waring

and Wells, 2015), therefore lights were turned off for this simulation. Following a similar approach to the experiment of Fry et al. (2011), the effect of $NO_3$ in the presence of ozone ($O_3$) is replicated by first injecting nitrogen dioxide ($NO_2$), limonene and carbon monoxide into the chamber, with the latter removing the effect of the hydroxyl radical - this initial injection of components marks the start of the experiment. After 1.5 hours $O_3$ is injected to initiate oxidation and in the absence of seed particles Fry et al. (2011) show that this mixture initiates a nucleation event, which we simulate here through the nucleation

parameterisation described below. Another injection of $O_3$, $NO_2$ and limonene at 4 hours with the addition of seed aerosol simulates indoor environments with substantial existing particulate matter.

    In Fig. 1 the particle organic nitrate curve demonstrates that around 15 % of SOPM comprises organonitrates that result from $NO_3$ reaction. Relative humidity was set to 50 % and water partitioning with particles is modelled using the gas-particle partitioning calculations described below. The particle inorganic nitrate curve represents the contribution of dinitrogen pentox-

ide ($N_2O_5$) to the particle-phase in PyCHAM. Gas-particle partitioning of $N_2O_5$ is driven by near-instantaneous hydrolysis to nitrate ions on contact with the aqueous phase of particles, which maintains a zero particle-phase mole fraction of $N_2O_5$. Thus, mass transfer of $N_2O_5$ to the particle continues according to the Raoult's law partitioning theory that we detail in Section 6. Here we simulate the instantaneous hydrolysis of $N_2O_5$ to nitrate ions by setting the activity coefficient to zero and estimating the accommodation coefficient according to Lowe et al. (2015), which estimates the rate of condensation as dependent on the

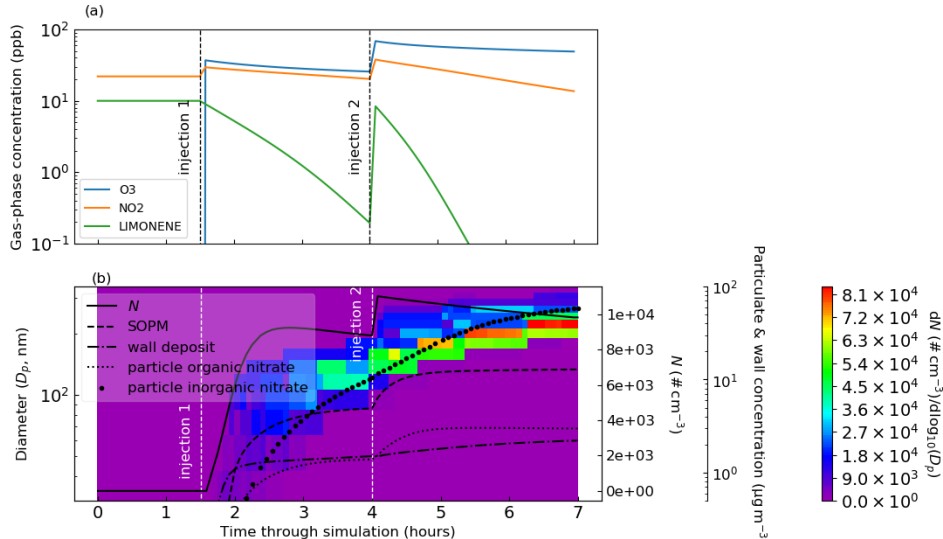

**Figure 1.** Limonene oxidation in the dark with and without seed particles. In (a), the gas-phase concentrations of key components and in (b), the particle properties. At the start 10 ppb limonene, 22 ppb $NO_2$ and 500 ppm CO are introduced. At 1.5 hours 38 ppb $O_3$ and 8 ppb $NO_2$ are injected (injection 1). At 4 hours a further injection of $O_3$ (45 ppb), limonene (10 ppb) and $NO_2$ (19 ppb) is coincident with an injection of seed aerosol (10 µg m$^{-3}$ with a mean diameter of 0.2 µm) (injection 2). In (b), the total particle number concentration ($N$) corresponds to the the first of the right axes, mass concentrations of: secondary organic particulate matter (SOPM), components deposited to walls (wall deposit), sum of particulate organic components with a nitrate functional group (particle organic nitrate) and sum of particulate inorganic components with a nitrate functional group (particle inorganic nitrate) correspond to the second of the right axes. Number size concentrations correspond to the filled contours, colour bar and left axis. Whilst $N$ includes seed particles, SOPM excludes the seed material and all mass concentrations exclude water.

Henry's law constant and diffusion constant of $N_2O_5$ in the aqueous phase. Assuming homogenous particles for a given size bin ($j$), Eq. 7 of Lowe et al. (2015) is:

$$\alpha_{N_2O_5,j,t} = \frac{4RT(0.03H_{N_2O_5,aq}D_{N_2O_5,aq})}{c_{N_2O_5}r_{j,t}}, \tag{1}$$

where $t$ represents time, $R$ is the universal gas constant, $T$ is temperature (set to 298 K), $c$ is average velocity in the gas phase and $r$ is particle radius. $H_{aq}D_{aq}$ is the product of the Henry's law constant and diffusion constant in the aqueous phase

for $N_2O_5$, with the former set to $5 \times 10^3$ M atm$^{-1}$ and the latter to $1 \times 10^9$ m$^2$ s$^{-1}$ (Lowe et al., 2015).

A current limitation of PyCHAM is its lack of explicit treatment of particle-phase processes including reaction and dissolution. Furthermore, there is currently no thermodynamic module to estimate divergences from ideality. As a result of these two limits, particle-phase processes can currently only be reproduced through manual setting of individual component activity

coefficients, accommodation coefficients and vapour pressures. For the example of Fig. 1, $N_2O_5$ is the only component to be treated non-ideally by setting of its activity coefficient to zero to simulate near-instantaneous hydrolysis to nitrate ions.

Studies have recently revealed the role of highly oxidised molecules (HOM) (Ehn et al., 2014), with the Peroxy Radical Autoxidation Mechanism (PRAM) simulating their chemistry (Roldin et al., 2019). For the results in Fig. 1, the PRAM scheme has been coupled with that of the Master Chemical Mechanism (MCM) (Jenkin et al., 1997; Saunders et al., 2003). It is intended that simulations such as Fig. 1 can help constrain uncertainties in new chemical schemes through comparison with observed particulate loading and composition analysis.

The primary difference between multiphase processes in simulation chambers and the real atmosphere is the presence of walls. Accurate representation of these processes in chamber models requires reasonable and realistic representation of the chamber walls (e.g Matsunaga and Ziemann, 2010; Zhang et al., 2015). Clearly PyCHAM must reasonably capture the partitioning of components and deposition of particles to chamber walls as incorrect reproduction makes comparison against measurements misleading or impossible. However, with correct wall loss constraint, simulations such as Fig. 1 can be compared against observations for verifying process understanding.

## 3  PyCHAM structure

For ease of navigation, PyCHAM has a modular structure with each key physicochemical process assigned an individual module. Unit tests are provided for modules, allowing the user to check a particular process is functioning correctly. It is intended that these tests will be useful for troubleshooting and for analysing the effects of modifications to modules.

At the core of PyCHAM lies simultaneous numerical integration of three coupled processes: gas-phase photochemistry, gas-particle partitioning and gas-wall partitioning. The units for rate of change for all processes is $\#\,\text{molecules}\,\text{cm}^{-3}\,\text{s}^{-1}$ and the Jacobian matrix is specified at each integration step. The ordinary differential equations (ODEs) for these processes are solved by the backward differentiation formula, for which utility has been demonstrated in similar atmospheric applications (Jacobson, 2005), from the CVODE Sundials software (Hindmarsh et al., 2005). We use Assimulo (Andersson et al., 2015), a python wrapper for sundials allowing communication between the solver and Python code.

The model structure is outlined in the schematic of Fig. 2, where we introduce the 'update time interval', which is an important input set by the user. This interval determines the frequency at which natural light flux is updated since it will change during the course of an open-roof experiment (in contrast to chamber lamps that have a constant actinic flux). The update time interval also sets the frequency at which particle number concentration is affected by coagulation, particle deposition to wall and nucleation (important for simulations involving particles). Particle number concentration and photolysis rates are constants in the ODEs for gas-particle partitioning and gas-phase photochemistry, respectively. The update time interval is passed to the integrator, which adaptively sets sub-time steps depending on problem stiffness. As such, a simulation without varying natural light and without particles could have an update time interval equal to the total experiment time without any loss of detail. For discontinuous changes to chamber conditions, such as turning lamps on/off, injection of gas-phase components or seed particles, automatic adaption of the update time interval ensures occurrence at the start of an integration step.

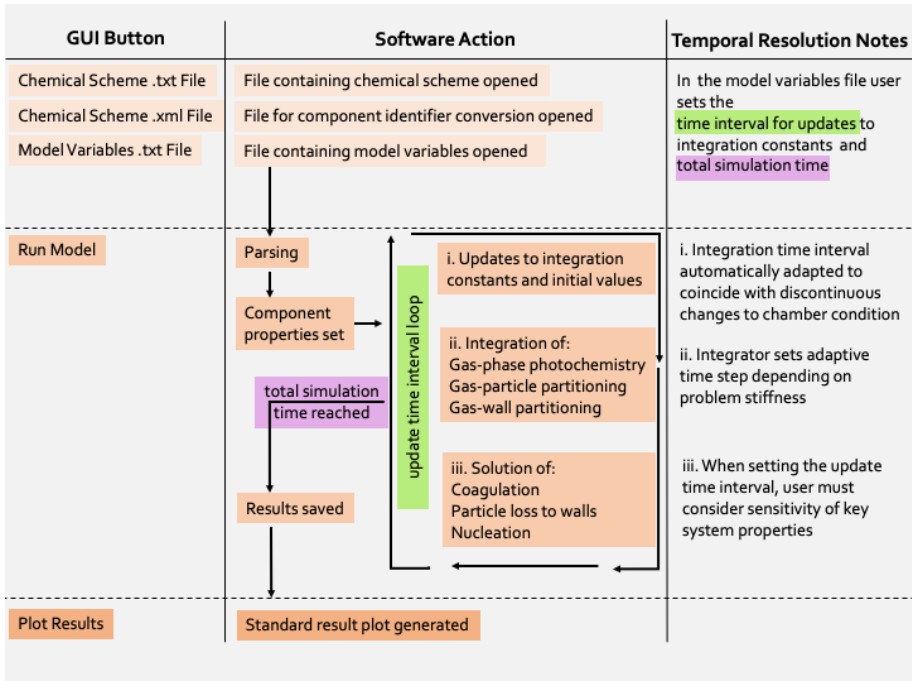

**Figure 2.** Schematic outlining the PyCHAM structure. In the left column are the GUI buttons that initiate the action(s) in the central column. On pressing the 'Run Model' button PyCHAM loops over the 'update time interval' until the experiment end time is reached. On each loop PyCHAM first checks whether any discontinuous changes to the chamber condition occur during the proposed time interval, and automatically reduces the interval if confirmed, such that the change can be implemented at the correct time at the start of the subsequent interval. Depending on problem stiffness the integrator uses sub-time steps, with the model taking the result at the end of the integration period. The final stage of a loop where the update time interval has been reached is solution of coagulation, particle deposition to walls and nucleation, which update the particle number concentration. The right column notes these temporal resolution aspects, with the mentioned sensitivity of key system properties investigated in Section 11.

Whilst coagulation and particle loss to wall have timescales of minutes to hours, nucleation can cause substantial changes within seconds (Section 11). Users may increase the update time interval, which has the advantage of decreased processing time, defined here as the time taken by a computer to complete all core PyCHAM commands, and the disadvantage of di-

vergence from high resolution estimates. Simulation sensitivity to temporal resolution is investigated in Section 11, including recommendations for maximum time intervals.

In the example model output of Fig. 1 several features of PyCHAM are demonstrated. First, the coupling of gas-phase chemistry and resulting partitioning of vapours with sufficiently low volatility to particles and walls. Nucleation has been simulated prior to the introduction of seed particles with values provided for the nucleation parameters (Section 10) that ensure

nucleation begins at the introduction of ozone, has a duration of thirty minutes and produces a peak number concentration

similar to that of the seed particle. Finally, coagulation and particle wall loss (the latter using the model of McMurry and Rader (1985)), contribute to the decay in particle number concentration.

The PyCHAM software is initiated via the command line to generate a graphical user interface (GUI). Via the GUI, users select three files (Fig. 2) representing: i) the chemical scheme, ii) a file associating the chemical identifiers inside the chemical scheme to their Simplified Molecular Input Line Entry System (SMILE) strings (Weininger, 1988), and iii) a model variables file. A fourth button on the GUI starts the simulation.

A parsing module interprets the chemical scheme and uses the chemical identifier conversion file to match component identifiers to their SMILE strings. For the chemical scheme file delimiting markers are required at certain points, however the form of these markers and the structure of the chemical scheme file may vary so long as the markers are specified by the user in the model variables file before running the programme. Consequently, a chemical scheme file downloaded directly from the MCM website may be used without modification in combination with a relevant model variables file.

On running PyCHAM, modules are automatically created that will track chemical tendencies (rate of change due to individual chemical reactions) and process tendencies (rate of change due to gas-particle and gas-wall partitioning) of specified components. A gas-phase initiation module interprets the user-defined starting concentrations of components, whilst a particle-phase initiation module establishes any seed particles at experiment start. The integration module is then called where the ODEs for gas-phase photochemistry and partitioning are solved (Fig. 2). A saving module stores results by default for gas and wall concentrations of all components, corresponding time and constants such as component molecular weight. If the user has setup the simulation to include particles, then particulate concentration for all components and particle number size distributions (with and without water) are also saved. If the user has defined components to track, then the change tendencies of these are saved, including rates of change due to photochemistry and partitioning to particles and walls.

The fifth button on the GUI will display and save graphs of the temporal profiles of number size distribution, secondary aerosol mass concentration, total particle number concentration, and the gas-phase concentrations of the components whose initial concentrations are user-defined. Besides these default plots, utility is enhanced by additional examples of plotting scripts (those used for figures here) in the PyCHAM repository. Furthermore, the script vol_contr_analys provides an example of how to generate a comma separated value (csv) file containing the names of components, their saturation vapour pressures at 298.15 K and particle-phase mass concentrations as a function of time. This script can also plot a volatility basis set analysis of particle-phase mass fraction with time. The programme can be stopped via the terminal when in integration mode, or outside this mode it can be terminated by closing the GUI.

Below we describe and verify the processes described above as coded in PyCHAM. Necessarily each process is examined in isolation, however, Fig. 1 and its associated text illustrate the coupling of mechanisms for a real world application.

## 4 Model variables and component properties

As described above, to initiate PyCHAM the user selects a completed model variables input file (an example is provided with the software). The available variables are extensive to allow adaptability to a range of experiments, consequently for a given

experiment, many of the variables in this file may be left empty. Here we introduce the available variables, whilst details such

as default values and units are provided in the appendix Table A. Several model variables (PyCHAM names of variables given in brackets) are purely functional, these include the name of the output file (res_file_name), whether to update the component property estimation files (umansysprop_update), which are described below in this section, the markers used to separate sections of the chemical scheme (chem_scheme_markers), names of files containing actinic flux (act_flux_file) and absorption cross-sections and quantum yields for photochemistry (photo_par_file). Total simulation time (total_model_time), the time

interval for updating integration constants (update_step) and the time interval for recording results (recording_time_step) are also available.

Chamber temperature (temperature) can change during a simulation by stating the corresponding time (tempt), whilst pressure (p_init) and relative humidity (rh) are also input. All relevant thermodynamic properties change accordingly with temperature, such as component vapour pressures and gas-phase diffusivities.

For simulations involving natural light, latitude (lat), longitude (lon), day of year (DayOfYear) and start time (daytime_start) are required inputs. Whether artificial or natural, users specify when (light_time) light is on or off (light_status). Any dilution rate (dil_fac) should be stated, or else the default is zero.

Initial concentrations (C0) of specified trace gases (Comp0) are stated separately to the concentrations (Ct) of specified trace gases (Compt) injected effectively instantaneously at set times (injectt) during the experiment. Specified components

(const_comp) will have a constant concentration for the entire experiment. The final option for introducing named components (const_infl) is to state the rate of their influx (Cinfl) during a set period of the experiment (const_infl_t). The change tendencies (defined above) of certain components (tracked_comp) can be recorded, which is helpful for analysis and troubleshooting.

For specific components (vol_Comp), liquid-phase saturation vapour pressures can be manually assigned (volP). As can activity coefficients (act_user) and accommodation coefficients (accom_coeff_user).

To simulate gas-wall partitioning, the mass transfer coefficient (mass_trans_coeff) and effective absorbing wall mass concentrations (eff_abs_wall_massC) can be set.

For the particle phase, users state the number of size bins (number_size_bins), size at lowermost size bin boundary (lower_part_size) and at uppermost boundary (upper_part_size), and whether to have linear or logarithmic spacing of size bins (space_mode). Setting size bin number to zero turns off particle considerations. As detailed below, users can also specify whether to use the

200 moving-centre or full-moving size structures for dealing with changing particle number size distributions (size_structure).

For seeded experiments, the component comprising the seed (seed_name), its molecular weight (seed_mw), density (seed_dens) and dissociation constant (core_diss) can be input. Either the particle concentration per size bin or the total particle concentration can be input (pconc), along with the time of particle injection (pconct). If the total particle concentration is given, this can be distributed across size bins by stating the mean radius (mean_rad) and standard deviation (std).

For nucleation experiments, the nucleating component (nuc_comp) can be changed from the default, as can the radius of newly nucleated particles (new_partr). To specify the temporal profile of nucleation, three parameters (detailed in Section 10) are input (nucv1, nucv2, nucv3).

Coagulation (coag_on) and particle loss to wall (McMurry_flag) can be turned off and on. If the latter is turned on, users can specify the size-dependent loss to walls (inflectDp, Grad_pre_inflect, Grad_post_inflect and Rate_at_inflect), or can invoke the McMurry and Rader (1985) model by also inputting the chamber wall surface area (Cham_SA), the charge per particle (part_charge_num) and the chamber electric field (elec_field), which are detailed in Section 9.

The components included in the user-defined chemical scheme are automatically allocated three properties by the PyCHAM software: molecular weight, pure component liquid density and pure component liquid saturation vapour pressure. Molecular weights are estimated by passing SMILE strings to the pybel module of the Open Babel chemical toolbox (O'Boyle et al., 2011). Open Babel is installed as part of the PyCHAM package and generates unique chemical identifiers for each component based on their SMILE string. For estimating component densities and liquid-phase saturation vapour pressures, the pybel chemical identifiers are passed to the UManSysProp module (Topping et al., 2016) which is updated on the first run of PyCHAM and at the request of the user (via the model variables file) thereafter (requires internet connection). By default the UMansSysProp module applies the liquid density estimation method of Girolami (1994) (recommended by Barley et al. (2013)) and the liquid saturation vapour pressure estimation method of Nannoolal et al. (2008) (recommended by O'Meara et al. (2014)). Component vapour pressures have a first order effect on absorptive partitioning between phases, however estimates for components with relatively low vapour pressures (below 1 Pa at 298.15 K) are associated with considerable uncertainty (O'Meara et al., 2014), as these are most difficult to measure experimentally and therefore inform estimation methods. Consequently, users can also specify the vapour pressures of certain components. Similarly, although the default activity and accommodation coefficient for all components partitioning to particles and wall is unity, users may set an alternative value for specific components. At present, activity coefficient calculations are not incorporated into PyCHAM.

## 5 Gas-phase photochemistry

For a chamber experiment including injection of reactive components, chemical reactions in the gas-phase drive the disequilibria that can affect the composition of gas, particle and wall. As mentioned above, schemes such as the MCM provide near-explicit gas-phase chemistry mechanisms for numerous organic precursors, and developments such as PRAM (Roldin et al., 2019) can be used to provide supplementary detailed updates to our understanding of atmospheric chemistry. PyCHAM is designed to accommodate any such detailed chemical schemes whilst also accepting very simplified or even empty (e.g. for a control simulation comprising only seed particles) chemical equation files. Whilst the software manual details the requirements for input chemical schemes and chemical identifier conversion files, here we describe how PyCHAM deals with chemistry. Equations of the general form:

$$s_{r_1} r_1 + s_{r_2} r_2 \ldots \rightarrow s_{p_1} p_1 + s_{p_2} p_2 \ldots \tag{R1}$$

where $s$ represents stoichiometric number, $r$ reactants and $p$ products, are expressed as the ODEs:

$$\frac{d[r_i]}{dt} = -s_{r_i} k_r \Pi_{j=1}^n \left( [r_j]^{s_{r_j}} \right) \tag{2}$$

$$\frac{d[p_i]}{dt} = s_{p_i} k_r \Pi_{j=1}^n \left( [r_j]^{s_{r_j}} \right) \tag{3}$$

where $n$ is the total number of reactants and $r_j$ is a given reactant for a given reaction. $k_r$ is the reaction rate coefficient.

For simulations involving gas-phase chemistry, users must therefore provide a reaction(s) of the form in Eq. R1 and an associated reaction rate coefficient inside a chemical scheme file. Naming of chemical components inside the chemical scheme is unrestricted, however, the software must be able to convert names to SMILES (Weininger, 1988). Therefore, users must provide a separate file stating a unique SMILES string for every component (Fig. 2). Examples of both the chemical scheme

and SMILES string conversion file are included in the software.

Inside the parsing module, reaction rate coefficients, reactant and product identities and their stoichiometric numbers are established from the chemical scheme file. To separate these properties either default formatting may be used, or a variant, so long as the appropriate changes are made inside the model variables file. By default, MCM Kinetic PreProcessor (Sander and Sandu, 2006) formatting is used (Jenkin et al., 1997; Saunders et al., 2003) and PyCHAM has been rigorously tested using

schemes and SMILE conversion files from the MCM website (Rickard and Young, 2020).

Reaction rate coefficients can be functions of temperature, relative humidity, pressure and concentrations of: third body, nitrogen, oxygen and peroxy radicals. Third body, nitrogen and oxygen concentrations are calculated by the ideal gas law with the user-set temperature and pressure. As in the MCM, the chemical scheme file can include generic reaction rate coefficients (those that have an identifier which is used as the reaction rate coefficient for one or more reactions).

Photochemistry is controlled through stating light on/off times inside the model variables file. The treatment of photochemistry is determined by the user and depends on the chemical scheme employed. In the case of the MCM scheme and natural sunlight, the scattering model based on Hayman (1997) and described in Saunders et al. (2003) is invoked by stating the relevant spatial and temporal coordinates in the model variables file. For artificial lights, users must provide a file stating the wavelength-dependent actinic flux (as described in the manual). The model then calls on either the absorption cross-section

and quantum yield estimates of MCM v3.3.1 or of a user-defined file.

## 5.1 Assessment of gas-phase photochemistry accuracy

To assess the accuracy of the photochemistry section of PyCHAM, gas-particle partitioning and gas-wall partitioning were turned off, leaving only gas-phase chemistry to be solved. Here we compare against AtChem2 (Sommariva et al., 2018) as a model benchmark, with both using MCM chemical schemes. Figure 3 shows the deviation with experiment time for two

standard aerosol chamber characterisation experiments: $\alpha-$pinene ozonolysis in the presence (plot a) and absence (plot b) of $NO_x$. To test both dark and illuminated scenarios, the simulation is for an aerosol chamber with an open roof, starting at midnight and finishing at midday. Initial concentrations of $\alpha-$pinene and $O_3$ were equal at 21.1 ppb for both experiments, whilst for $NO_x$ the initial concentration was 9.8 ppb in Fig. 3a and 0 ppb in Fig. 3b. Latitude was set to 51.51, longitude to 0.13 (London, UK) and the date to 1 July. In both models, absolute error tolerance was set to $1 \times 10^{-3}$ and relative tolerance was set

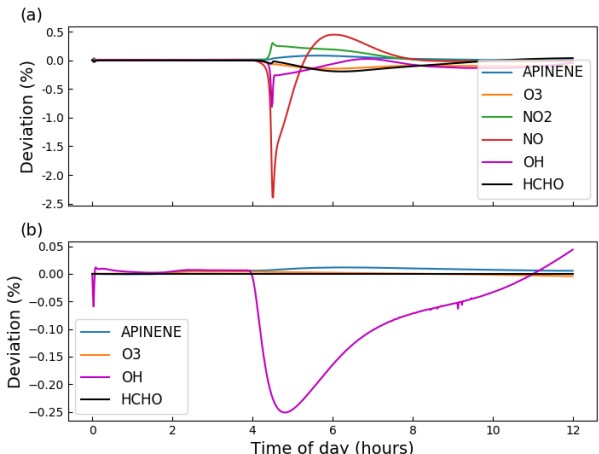

**Figure 3.** Gas-phase photochemistry verified; simulations of photochemistry in an aerosol chamber exposed to natural light, where deviation is defined in Eq. 4. $\alpha-$pinene ozonolysis is simulated in both plots, with $\alpha-$pinene and $O_3$ given the same initial concentrations of 21.1 ppb, and initial $NO_x$ concentration in (a) 9.8 ppb and in (b) 0 ppb. For both simulations, the aerosol chamber is transparent and exposed to daylight without cloud interference, with dawn at approximately 4 hours. Gas-wall partitioning is turned off in PyCHAM to be consistent with the AtChem2 model and no particles are present in either model.

to $1 \times 10^{-4}$. AtChem2 has no functionality for gas-particle partitioning which is fine for this section dealing with only gas-phase photochemistry, whilst PyCHAM had particle considerations turned off. The deviation between PyCHAM and AtChem2 was calculated using:

$$\sigma_{i,t} = \left( \frac{s_{i,t} - b_{i,t}}{\vee(b_i)} \right) 100, \tag{4}$$

where $\sigma_{i,t}$ is the percentage deviation (%) for component $i$ at time $t$, $s$ is the PyCHAM result, $b$ is the AtChem2 result. $\vee(b_i)$

is the AtChem2 maximum for a given component during the simulation which is the chosen scaling factor for deviations as it means any difference between model estimates is referenced against a reasonable value for that component (in contrast scaling by $b_{i,t}$ when $b_{i,t} \ll \vee(b_i)$ may introduce a very large percentage deviation for a relatively very small difference between model estimates).

   Whilst Fig. 3 indicates that PyCHAM performs well for components with both relatively short (e.g. OH) and long (e.g.

$\alpha-$pinene) lifetimes, it is necessary to ascertain that agreement is gained through the correct mechanism. The chemical change tendencies of formaldehyde were tracked in both PyCHAM and AtChem2 for the $\alpha-$pinene ozonolysis simulations used for Fig. 3. Deviations of PyCHAM results from AtChem2 were calculated using Eq. 4, but with concentrations replaced by change tendencies resulting from individual reaction channels. Of the loss and production channels for formaldehyde the two of each

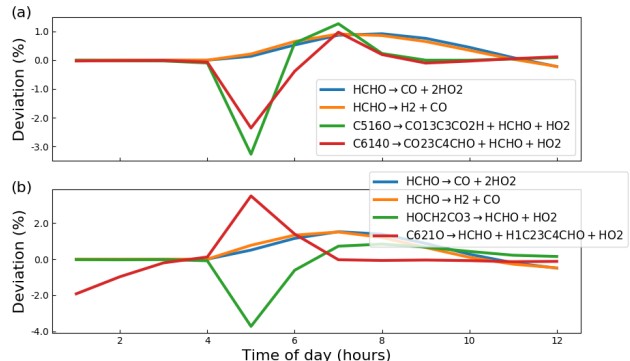

**Figure 4.** Deviation of PyCHAM change tendency (rate of change due to individual chemical reactions) for formaldehyde (HCHO) from AtChem2 simulations for the MCM reactions given in the legends (the loss and production channels of formaldehyde with greatest deviation). Where the definition for deviation is given by Eq. 4. Both plots are results for the $\alpha-$pinene ozonolysis reaction described in the main text for Fig. 3, with (a) in the presence of $NO_x$ and (b) in the absence of $NO_x$.

with greatest deviation are shown in Fig. 4. The low deviation values in Fig. 4 demonstrate that PyCHAM indeed solves
gas-phase photochemistry correctly.

## 5.2   PyCHAM sensitivity to temporal resolution of continuous photolysis change

For minimising processing time, users can increase the update time interval (Fig. 2), however this decreases the frequency of update for natural light intensity (note that for artificial light simulations, PyCHAM automatically adapts the time interval to coincide with timings of lights being turned on or off). For open roof experiments, increasing the update time interval therefore
reduces the accuracy of estimated photolysis rates. To illustrate and quantify the issue, the same scenario described above for Fig. 3 is used, i.e. gas-phase chemistry only simulation with increasing natural sunlight intensity. Now we compare PyCHAM low temporal resolution (update time intervals of $6x10^2$ and $6x10^3$ s) with PyCHAM high resolution (updates every $6x10^1$ s). To quantify divergence of low resolution results from high we use Eq. 4. Fig. 5 shows the loss of accuracy rising to 20 % for the lowest resolution case for both short- and long-lived components. Processing time for the $6x10^3$ s resolution was 52 s using a
2.5 GHz Intel Core i5 processor; this increased by a factor of 7 for the $6x10^2$ s resolution and by 53 for the $6x10^1$ s resolution. Users should conduct a similar test if their chemical scheme or environmental conditions vary significantly from those here.

## 6   Gas-particle partitioning and sectional approach

PyCHAM simulations, like chamber experiments, are possible with and without seed particles. For seed particles, the user defines number size distribution and composition inside the model variables input file. Furthermore, because PyCHAM uses

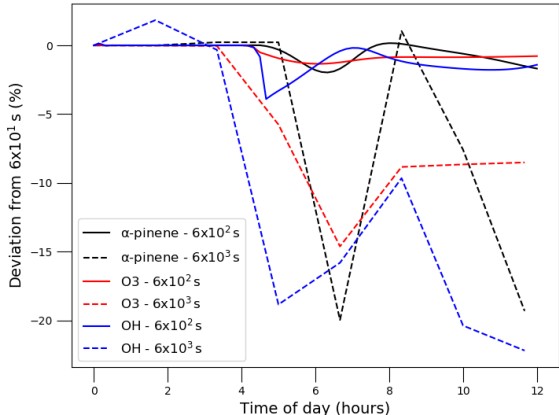

**Figure 5.** Illustrating the effect of update time interval resolution in PyCHAM on gas-phase concentrations of $\alpha-$pinene, $O_3$ and OH for the $\alpha-$pinene ozonolysis in presence of $NO_x$ experiment described above for Fig. 3. Time intervals were set to $6x10^2$ and $6x10^3$ s as shown in the legend, and the deviation is that from results for a time interval of $6x10^1$ s.

size bins to discretise particles, users can state the number of size bins, lower and upper bin bounds and whether to use linear or logarithmic spacing of size bins.

Particle number size distribution can change as a direct consequence of four processes modelled by PyCHAM: gas-particle partitioning, coagulation, nucleation and particle loss to walls. Whilst coagulation, nucleation and particle wall loss are discussed below, gas-particle partitioning is solved with (Zaveri et al., 2008):

$$\frac{dC_{i,g}}{dt} = -\sum_{j=1}^{N} k_{i,j}(C_{i,g} - x_{i,j}p_i^0 K_{v,j}\gamma_{i,j}), \tag{5}$$

$$\frac{dC_{i,j}}{dt} = k_{i,j}(C_{i,g} - x_{i,j}p_i^0 K_{v,j}\gamma_i), \tag{6}$$

where component $i$ is partitioning to a size bin $j$ from the gas $g$, with $N$ size bins present. $x$ is the particle-phase mole fraction, $p^0$ is the pure component liquid (sub-cooled if necessary) vapour pressure, $K_v$ is the Kelvin factor and $\gamma$ is the activity coefficient. $k_{i,j}$ is the first order mass transfer coefficient for component $i$ to size bin $j$, which includes the transition regime correction factor of Fuchs and Sutugin (1971). As suggested by Eq. 5 all components are allowed to partition between the gas and particle phase, with the concentrations of individual components in each size bin of the particle phase tracked in addition to their gas-phase concentrations (both in units of $molecules\,cm^{-3}$). The unit test test_partit_var is available to check that the Kelvin and Raoult effects of the PyCHAM gas-particle partitioning equation are accurate (e.g. through comparison with Fig. 16.1 of Jacobson (2005)). By default PyCHAM assumes an ideal system with no particle-phase mass transfer limitation to partitioning. However, users can diverge from ideality through setting of $\gamma$ and they may replicate changes in particle viscosity via the accommodation coefficient, which is used to estimate $k_{i,j}$.

In this section we focus on how PyCHAM treats changing particle number size distribution due predominantly to gas-particle partitioning. PyCHAM provides the option of using either the moving-centre or full-moving size structure (Jacobson, 2005) for evolving size distributions. The moving-centre approach has the advantage of minimal numerical diffusion and the readily

accommodates populations of particles of varying modes (e.g. a nucleation event in the presence of pre-existing particles). However, it suffers from loss of accuracy due to averaging of particles originally from different size bins that have grown, shrunk or coagulated to a given size bin (Zhang et al., 1999). In contrast the full-moving structure does not average particles of different size bins together following gas-particle partitioning, and can therefore exactly model certain chamber scenarios. The full-moving structure less readily accommodates multiple populations of particle, such as the nucleation event followed by seed

particle injection used in Fig. 1. Therefore, in the interest of generality the default size structure in PyCHAM is moving-centre and unless otherwise stated all results in this paper use it.

Here we assess the moving-centre and full-moving size structures through analysis of output during two periods of relatively substantial (and therefore testing) condensational growth and compare to benchmark simulations. The simulations also illustrate two further means of component influx to chambers using PyCHAM in addition to the simulations above where

components were introduced with an initial pulse. In the first case a constant flux of sulphuric acid is added to a chamber with seed aerosol typical of hazy conditions following the benchmark simulation of Zhang et al. (1999). For consistency with the benchmark, gas and particle partitioning to walls was turned off and sulphuric acid was assumed to be non-volatile. The analysis section of Zhang et al. (1999) notes that to resolve the growth of smallest particles in this scenario, spatial resolution must be at least 100 size bins, therefore we use this value and set the update time interval to 90 s for a total 12 hour simulation.

The exact solution to this condensational growth problem is given by the full-moving output in Fig. 6a, which replicates that in Fig. 3 of Zhang et al. (1999). Comparing the moving-centre output against the full-moving we see that the tri-modal distribution is present with mean values at the correct particle size though with lower peak height and greater spread (for both volume and number size distributions), indicating numerical diffusion. The degree of agreement is significantly better than for the 13 size bin moving-centre simulation presented in Zhang et al. (1999) and indicates that PyCHAM is operating as intended.

Our results in Fig. 6a are a two-point moving average which is often necessary for the moving-centre structure because of its requirement that all particles in a size bin be transferred to the adjacent bin, meaning that some bins will intermittently have zero particles.

Another case of relatively intense vapour-particle partitioning is provided by the example of cloud condensation nuclei experiencing varying degrees of water vapour supersaturation. Chamber experiments may involve injections of a component at

specific times and the model variables input file can accommodate such a scenario. Making use of this function we reproduce the benchmark simulation of Jacobson (2005) (Fig. 13.8) where relative humidity is increased to 100.002 % every minute (including at simulation start) for nine minutes, with results analysed after ten minutes. Seed particles are assumed non-volatile and wall interactions are turned off. The parameters: temperature, seed component dissociation constant, molecular weight and density are not disclosed by the reference simulation, therefore we set these as: 300 K, 1, 350 $\mathrm{g\,mol^{-1}}$ and 0.9 $\mathrm{g\,cm^{-3}}$,

respectively. The comparison between the Jacobson (2005) result in Fig. 6b and PyCHAM outputs certainly shows agreement in the main feature of this simulation, which is the initially larger particles out competing smaller particles for water condensation

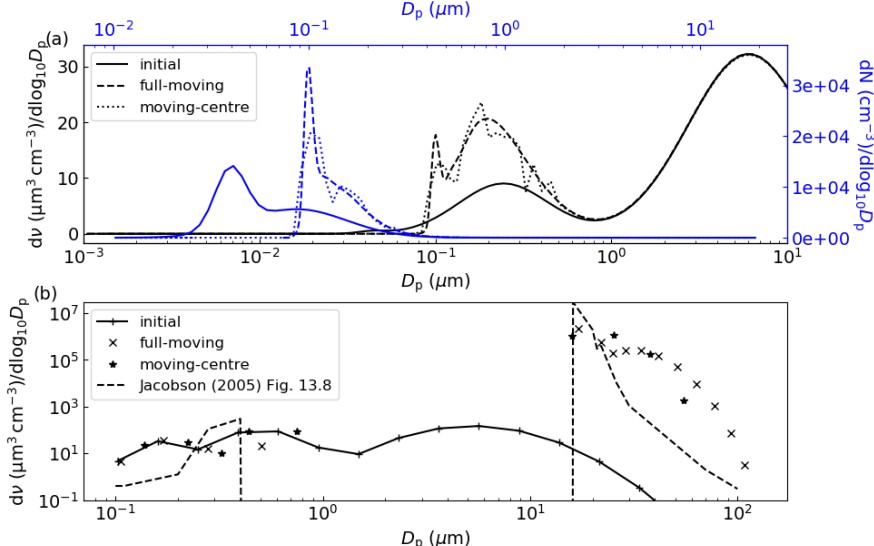

**Figure 6.** In (a), replication of Fig. 3 of Zhang et al. (1999) where a constant influx of sulphuric acid condenses to seed particles with the shown initial volume size distribution (bottom and left axis, in black) and initial number size distribution (top and right axis, in blue), with final results after 12 hours. In (b), replication of Fig. 13.8 of Jacobson (2005), where an initial distribution of particles are subject to a relative humidity of 100.002 % at minute intervals for 9 minutes, with results shown after 10 minutes.

to grow to water droplet size ($D_p > 10$ µm). It should be noted that this is a very much more stringent test of the representation of partitioning than is ever intended for PyCHAM, which will not generally be used for the huge mass flux of condensing material experienced under water supersaturated conditions. Nevertheless, the PyCHAM result gives reasonable agreement

considering that key parameters (such as seed component dissociation) may vary between simulations and taken together with Fig. 6a verifies the operation of gas-particle partitioning and both the moving-centre and full-moving size structure.

## 7  Gas-wall partitioning

The partitioning of gases to the chamber wall is often termed wall loss as the net movement is from the gas phase to the wall (for an initially clean chamber wall). Traditionally this process has been viewed as an inconvenience since chamber results

often depend on the concentration of gas- and particle-phases of certain components, whilst the fraction of these components lost to walls is poorly constrained. Several studies have focussed on partitioning to Teflon walls, which are frequently employed (Matsunaga and Ziemann, 2010; Zhang et al., 2015; Zhao et al., 2018), however, the process remains poorly modelled across the wide range of chamber materials, relative humidities, gas-phase loading, component volatilities and activity coefficients present in chamber experiments (e.g. Stefenelli et al., 2018).

A volatility-dependent gas-wall partitioning parameterisation for Teflon chambers has been suggested by Krechmer et al. (2016), however they recommend further investigation into its suitability across chamber experiment conditions. Because this paper is focussed on describing PyCHAM in its most generally applicable form we do not analyse specific models here, however, users are encouraged to adapt the programme based on their particular experiment conditions.

     It is therefore preferable to allow the user to fit vapour losses to walls through the tuning of two wall loss parameters, one
primarily determining equilibrium, called the effective wall mass concentration ($C_w$), and one determining rate of partitioning, the mass transfer coefficient ($k_w$). These influence gas-wall partitioning through an equation of the same framework as gas-particle partitioning (which is described in Section 6 and in Zaveri et al. (2008)):

$$\frac{dC_{i,g}}{dt} = -k_w(C_{i,g} - \frac{C_{i,w}}{C_w}p_i^0\gamma_i), \tag{7}$$

$$\frac{dC_{i,w}}{dt} = k_w(C_{i,g} - \frac{C_{i,w}}{C_w}p_i^0\gamma_i), \tag{8}$$

where $p_i^0$ is the liquid (sub-cooled if necessary) saturation vapour pressure of component $i$ and $\gamma_i$ is its activity coefficient on the wall. Following the conclusions of Matsunaga and Ziemann (2010) and Zhang et al. (2015), $k_w$ represents factors such as gas- and wall-phase diffusion, turbulence, accommodation coefficient and the chamber surface area to volume ratio, whilst $C_w$ reflects the adsorbing and/or absorbing properties of the wall, including effects of relative humidity, surface area, diffusivity and porosity. We recommend the iterative fitting of $k_w$ and $C_w$ to observations through minimising observation-model residuals.
$C_w$ in PyCHAM does not vary with the mass transferred to the wall, which is consistent with the findings of Matsunaga and Ziemann (2010) and Zhang et al. (2015) that indicate the effective mass concentration of the wall is much larger than the mass concentration of transferred material.

## 7.1   Tuning gas-wall partitioning parameters

     Next we illustrate the sensitivity to $k_w$ and $C_w$ in Eq. 7. The same simulation setup described above for Fig. 3 was used
though with $\alpha-$pinene replaced by isoprene (using the chemical scheme of MCM v3.3.1) with a concentration at experiment start of 63.4 ppb. Seed particles comprised of ammonium sulphate with mean diameter $1\mathrm{x}10^{-1}$ μm and number concentration $1.5\mathrm{x}10^4$ cm$^{-3}$ were introduced at experiment start, this equates to a mass concentration of $8\,\mu\mathrm{g\,m}^{-3}$ . Pure component liquid saturation vapour pressures were estimated by the Nannoolal et al. (2008) method and activity coefficients for all components were assumed to be unity. In this example and throughout this paper, all components are allowed to partition between the gas
and particle phase according to Eq. 5.

     To begin, both $k_w$ and $C_w$ were set sufficiently low to effectively eliminate gas-wall partitioning. Second, $C_w$ was set to $1\mathrm{x}10^{-4}\,\mu\mathrm{g\,m}^{-3}$ and $k_w$ set to $1\mathrm{x}10^{-3}\,\mathrm{s}^{-1}$ at which a notable decrease in secondary particulate matter (SPM) was observed. Third, $k_w$ was held constant whilst $C_w$ was raised two orders of magnitude greater to $1\mathrm{x}10^{-2}\,\mu\mathrm{g\,m}^{-3}$. Fourth, $C_w$ was held constant at $1\mathrm{x}10^{-4}\,\mu\mathrm{g\,m}^{-3}$ and $k_w$ was raised by five orders of magnitude to $1\mathrm{x}10^2\,\mathrm{s}^{-1}$. The effect on secondary particulate
matter mass concentration is given in Fig. 7 and demonstrates that at sufficiently large values of $C_w$ and $k_w$, SPM production can be effectively suppressed through competitive uptake of vapours to chamber walls. However, it should be noted that for a

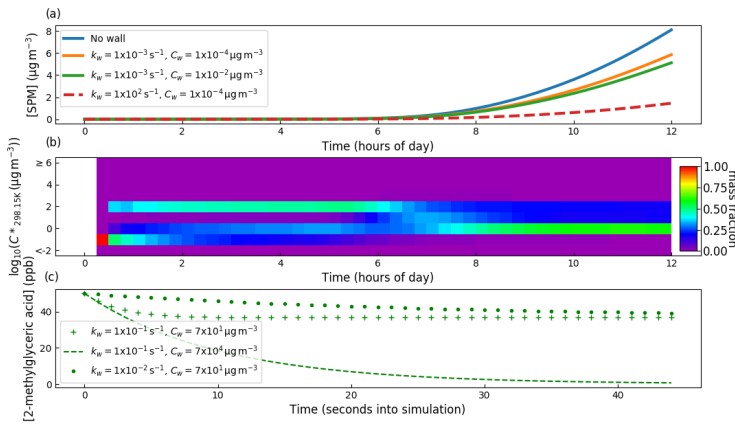

**Figure 7.** In (a) sensitivity of SPM mass concentration to the gas-wall partitioning parameters $k_w$ and $C_w$ from Eq. 7. Seed particles with a concentration of $8\,\mu g\,m^{-3}$ were present at the start of the experiment. Initial concentrations of $O_3$, isoprene and $NO_2$ were set to 21.1, 63.4 and 9.8 ppb, respectively. With regards to photolysis rates, the simulation made the same considerations as in Fig. 3, where natural sunlight drove reactions after dawn at approximately 4 hours. Plot (b) illustrates a PyCHAM plotting tool for contributions of components in volatility bands to secondary particulate matter as a function of time. The example given is for the No wall case of (a). In (c), sensitivity of the gas-phase concentration decay of 2-methylglyceric acid is shown for a control experiment where only a single organic component is present in the absence of particles. Wall loss parameters are given in the legend.

given $C_w$, there is a limit to suppression of SPM due to $k_w$ increase as it affects only the rate of partitioning with walls rather than the condensable fraction.

Whilst the intention of this section is demonstration of gas-wall partitioning, and it is beyond the scope of this paper to
evaluate the chemical schemes or property estimation techniques that may be used by PyCHAM, comparison of Fig. 7a with observations is insightful. As discussed earlier, currently PyCHAM does not account for heterogenous or particle-phase reaction or oligomerisation. However, these processes are reported to have a significant impact on secondary particulate matter formation from isoprene (Sato et al., 2011; D'Ambro et al., 2017), therefore we would expect a lower yield in Fig. 7a than reported from observational studies. Observations from a similar aerosol chamber experiment are presented in Table 1 (ex-
periment number 10) of Liu et al. (2016). Consistent with our expectation, after wall loss correction they report a secondary organic aerosol yield of 11.6 %, whereas for Fig. 7a here the No wall case gives a yield of 4.5 %. However, Sato et al. (2011) observe a yield of 5.5 % for isoprene photooxidation in the presence of 22 ppb $NO_x$ and neutral seed particles. Given that Sato et al. (2011) also observe up to 1/3 the organic particulate concentration comprising oligoesters that are not currently simulated in PyCHAM, the yield found here is relatively high. Whilst not in the scope of the current paper to pinpoint the cause of this
discrepancy, it is worthwhile noting that in Table 6 of O'Meara et al. (2014) the Nannoolal et al. (2008) estimation method for component vapour pressures generated between two to three times more secondary particulate matter mass concentration than when measured vapour pressures were used. This is due to the disproportionate effect on SPM from underestimating

component vapour pressures compared to overestimating (the Nannoolal et al. (2008) method was found to have a relatively low overall bias in O'Meara et al. (2014)).

In Fig. 7b is demonstrated a useful plotting tool (script name: vol_contr_analys) available to PyCHAM users for analysing the contribution of components in a volatility basis set (i.e. grouped into different volatility bands) to secondary particulate matter. This plot relates to the No wall simulation in Fig. 7a and indicates substantial changes to the mass fraction contributed by each volatility bin with time. Prior to dawn at approximately 4 hours, relatively low volatility components contribute significantly to SPM. Also using the PyCHAM script vol_contr_analys, a comma separated value file with the vapour pressures and mass concentrations of individual components in the particle-phase is produced. Analysis of this file shows that for the volatility band centred on $1x10^{-1.5}$ µg m$^{-3}$, the component with MCM name C536OOH dominates the mass fraction, though we note that checking the No wall curve in Fig. 7a, it's absolute contribution to SPM is negligible relative to later SPM values. C536OOH has three carboxylic acid groups and a ketone group. Whilst such highly functional components are expected to have relatively low vapour pressures, we advise readers of the high error associated with estimated vapour pressures for such components; Fig. 5 of O'Meara et al. (2014) demonstrates six orders of magnitude underestimation compared to measurements.

To guide constraint for wall loss parameters, we follow the example of Matsunaga and Ziemann (2010), with a control experiment comprising a single semi-volatile component introduced to the chamber at the start of the simulation at 50 ppb. 2-methylglyceric acid is selected as it has an estimated particle mass concentration saturation vapour pressure ($C^*$) of $1.15x10^2$ µg m$^{-3}$ at 298.15 K (the simulation temperature) and is an observed oxidation product of isoprene (Surratt et al., 2006). No other components or particles are introduced. With regards to designing a control experiment for tuning $C_w$ and $k_w$, the results shown in Fig. 7c demonstrate that a component with a $C^*$ close to the $C_w$ value has large sensitivity to $C_w$, thereby allowing greatest ease of tuning. Note, that this sensitivity can be utilised through varying chamber temperature (and therefore the $C^*$ of a component), or through using a component with different volatility. Furthermore, to discern the effect of $k_w$ a component with substantial partitioning to walls is required. When quantifying $k_w$ it is worthwhile considering the required precision, because above a certain value, no further effect on SPM concentration results.

## 8 Coagulation

Equations of coagulation kernels for Brownian diffusion, convective Brownian diffusion enhancement, gravitational collection, turbulent inertial motion, turbulent shear and Van der Waals collision were taken from Jacobson (2005). The unit test test_coag produces a plot of coagulation kernels that can be compared to Fig. 15.7 of Jacobson (2005) to verify accuracy. The number concentrations of newly coagulated particles are allocated to size bin $k$ if they fit within the bounding volumes and do not

include a particle originally from $k$; for combinations of size bins $j$ and $z$ the fraction of a newly coagulated particle contributing to size bin $k$ is:

$$fp_{z,j,k} = \begin{cases} 1 & Vb_{k,l} \leq (V_{j\neq k,t-h} + V_{z\neq k,t-h}) < Vb_{k,u} \\ 0 & j = k \vee z = k \\ 0 & (V_{j,t-h} + V_{z,t-h}) < Vb_{k,l} \\ 0 & (V_{j,t-h} + V_{z,t-h}) \geq Vb_{k,u} \end{cases} \tag{9}$$

where $V$ is single particle volume, $Vb_{k,l}$ and $Vb_{k,u}$ are lower and upper volume bounds, respectively, and $h$ is the time interval for coagulation to occur over. Particle number concentration in size bin $k$ decreases when coagulation of $k$ particles with those from any size bin produce a particle of size greater than the upper size bin bound of $k$. If however, coagulation produces a particle within size bin $k$, there is no loss of number concentration if coagulation is with a smaller size bin. If the self-coagulation of $k$ produces a particle in size bin $k$ (a scenario dependent on the size bin width) the change is half that if the coagulated particle fits a size bin larger than $k$:

$$fl_{k,z} = \begin{cases} 1 & (V_{k,t-h} + V_{j,t-h}) \geq Vb_{k,u} \\ 0.5 & (V_{k,t-h} + V_{k,t-h}) < Vb_{k,u} \\ 0 & (V_{k,t-h} + V_{j\neq k,t-h}) < Vb_{k,u} \end{cases} \tag{10}$$

The semiimplicit coagulation equation from Jacobson (2005) is then used to estimate the new number concentration per size bin ($N$ ($\# \, \mathrm{cm}^{-3}$)):

$$N_{k,t} = \frac{N_{k,t-h} + \frac{1}{2}h(\sum_{j=1}^{k}(\sum_{z=1}^{k-1} fp_{z,j,k}\beta_{z,j}N_{z,t}N_{j,t-h}))}{1 + h\sum_{j=1}^{A} fl_{k,j}\beta_{k,j}N_{j,t-h}}, \tag{11}$$

where $A$ is the total number of size bins. It is necessary to solve Eq. 11 in ascending order of size bins because for the smallest bin, which cannot gain particles through coagulation, the term involving $N_{z,t}$ is zero. This makes Eq. 11 explicit for the first size bin and allows estimates of $N_{z,t}$ for larger size bins.

The semiimplicit approach has the advantage of being positive-definite and non-iterative, making it unconditionally stable and with lower processing time than an implicit numerical approach. However, whilst conserving number, Eq. 11 is not mass-conserving, whereas the implicit treatment (Jacobson, 2005) conserves both number and mass. It the intention of the core development team to explore the feasibility of modelling with the implicit approach since guaranteed mass conservation is preferable. To estimate mass gained by size bin $k$ due to coagulation, the number fraction of the original number of particles per size bin represented by coagulating particles is found, with this fraction of the original mass per size bin moved to the new

size bin. Considering now gain in mass concentration rather than number concentration, the contribution from size bins smaller than $k$ that coagulate with $k$ must be considered, so:

$$fp_{z,j,k} = \begin{cases} 1 & Vb_{k,l} \leq (V_{j,t-h} + V_{z \neq k, t-h}) < Vb_{k,u} \\ 0 & z = k \\ 0 & (V_{j,t-h} + V_{z,t-h}) < Vb_{k,l} \\ 0 & (V_{j,t-h} + V_{z,t-h}) \geq Vb_{k,u} \end{cases} \tag{12}$$

Now, for size bin $k$, the gain in mass concentration ($C$) of component $i$ from smaller size bins ($z$) (based on Eq. 11 and Eq. 12) is:

$$\Delta C_{i,k,t} = \sum_{z=1}^{k-1} \frac{h(\sum_{j=1}^{k} fp_{z,j,k} \beta_{z,j} N_{z,t} N_{j,t-h})}{N_{z,t-h}} C_{i,z,t-h} \tag{13}$$

For the loss of mass concentration from size bin $k$, size bin pair fractions become:

$$fl_{k,j} = \begin{cases} 1 & (V_{k,t-h} + V_{j,t-h}) \geq Vb_{k,u} \\ 0 & (V_{k,t-h} + V_{k,t-h}) < Vb_{k,u} \\ 0 & (V_{k,t-h} + V_{j \neq k, t-h}) < Vb_{k,u} \end{cases} \tag{14}$$

Loss of mass concentration due to coagulation of particles in $k$ producing particles in larger size bins is then (based on Eq. 11 and Eq. 14):

$$\Delta C_{i,k,t} = -\left(1 - \frac{1}{1 + h \sum_{j=1}^{A} fl_{k,j} \beta_{k,j} N_{j,t-h}}\right) C_{i,k,t-h}, \tag{15}$$

Equations 10-15 imply that coagulation directly influences the particle number- and mass-size distributions. We now asses the sensitivity of the number size distribution and mass conservation to temporal resolution (represented by the time interval for updating coagulation) and number of size bins. A relatively complex initial distribution with four number modes is taken from ambient observations at Claremont, California on August 27, 1987 (Jacobson, 2005) and assumed to comprise non-volatile material. Results are presented for a six hour simulation in Fig. 8 where particle wall loss was turned off to allow clearer assessment of the coagulation sensitivity. In the top row of Fig. 8 no gas-phase chemistry was allowed, whilst in the bottom row, a single chemical reaction with reaction rate $5.6\text{x}10^{-17} \, \text{molec}^{-1}\text{s}^{-1}$ between $\alpha-$pinene and $O_3$ (both with initial concentrations 100 ppb) was modelled to produce a single low volatility product with saturation vapour pressure of $1\text{x}10^{-10} \, \text{Pa}$, whilst gas-wall partitioning was turned off. For the chemistry case, approximately 500 µg m$^{-3}$ of secondary material was formed, compared to 90 µg m$^{-3}$ of seed material. Columns in Fig. 8 are distinguished by the number of size bins as presented in the column titles, and within each plot temporal resolution is varied.

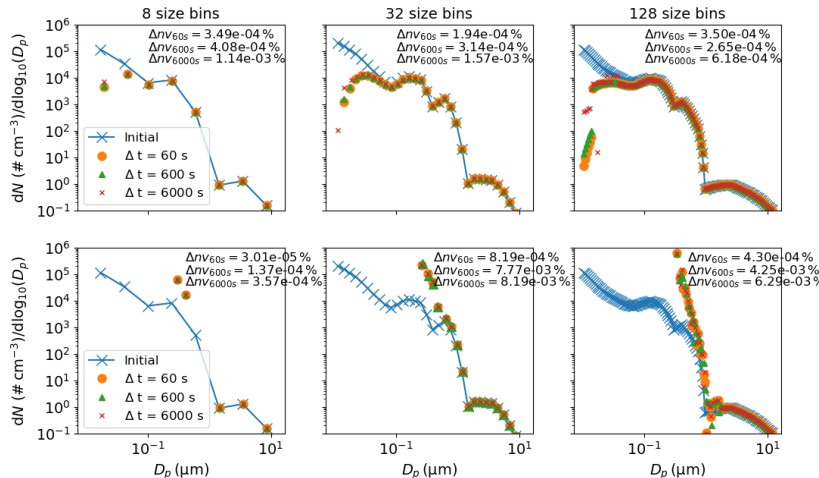

**Figure 8.** Sensitivity of the coagulation process to changes in temporal resolution (given in the legend) and number of size bins (given in column titles). In the top row no chemistry occurred whilst in the bottom row a semi-volatile species was produced, as detailed in the main text. Results are for the end of a simulated six hour experiment. The $\Delta nv_{\text{temporal resolution}}$ value given in the inset text is the percentage change in total non-volatile particle-phase material from the start to finish of the experiment, demonstrating mass conservation in the model.

The inset text of Fig. 8 ($\Delta nv$) gives the fractional change in non-volatile material from the start to end (six hours) of the simulation for the three temporal resolutions. It is clear that the coagulation equations introduce negligible error to mass conservation. Two features are present in the top row (no chemistry) of Fig. 8: first, that in terms of number concentration, coagulation overwhelmingly affects the number concentration of smaller particles - note that such particles are sufficiently small in volume that they may coagulate with a larger particle without causing it to grow a size bin; second, that only for the

smallest particles (below a diameter of $3\mathrm{x}10^{-2}$ $\mu$m in this case) is a sensitivity to temporal resolution clear across all size bin resolutions. For the no chemistry case there is demonstrable coupling of number of size bins and temporal resolution, with an increase in the former indicating greater sensitivity to the latter. However, the resolution considerations change substantially when we consider the case with gas-particle partitioning in the bottom row of Fig. 8. In this instance, the effect of partitioning dominates the change in number-size distribution and no sensitivity of coagulation to number of size bins or temporal resolution

is discernible. This is consistent with the typical timescales associated with the two processes: seconds to minutes for gas-particle partitioning and minutes to hours for coagulation (Seinfeld and Pandis, 2006). We recommend users consider these examples in addition to the nature of their simulation and objective when deciding whether temporal resolution or number of size bins will significantly impact results.

## 9   Particle deposition to walls

As with gas-wall partitioning, the loss of particles to chamber walls can significantly invalidate chamber results if unaccounted for and has been detailed in previous publications (Crump and Seinfeld, 1981; McMurry and Rader, 1985; Nah et al., 2017; Wang et al., 2018). During control experiments the deposition rate of particles to walls can be inferred through observations of the rate of decay of particles of varying size (with coagulation accounted for) (Charan et al., 2019). Several studies have published results from such experiments (McMurry and Rader, 1985; Wang et al., 2018), including a relatively large dataset from the EUROCHAMP2020 project (Oliveri, 2018). Comparison of inferred wall loss rates indicate that diffusion and settling enhance the loss rates of relatively small and relatively large particles, respectively (Crump and Seinfeld, 1981), however the absolute values and size-dependent gradient of the loss rates vary significantly between control experiments. Even for a given chamber, significant variations appear with changes to relative humidity, disturbance to walls due to air conditioning, and, for teflon chambers, with time since the chamber walls experienced frictional force to create electrostatic charge (Wang et al., 2018). Currently no method is available to measure the required inputs that a particle deposition model would need to satisfactorily reproduce observations across all chambers and conditions, therefore in PyCHAM users have three options to estimate particle wall deposition. Here we describe the options and provide examples of their use.

Users select wall loss treatment with the McMurry_flag option in the model variables input file. The default (if left empty) is no loss of particles to wall, which can be used for estimating wall loss corrected values such as aerosol yield. If set to one, the model of McMurry and Rader (1985) is used, which is based on the particle deposition model of Crump and Seinfeld (1981) but with electrostatic effects. Studies have found the Crump and Seinfeld (1981) and McMurry and Rader (1985) approach to reproduce measured particle wall losses well (Chen et al., 1992; Kim et al., 2001). Selecting McMurry and Rader (1985) requires the user to also input the chamber surface area, the average charge per particle and the average electric field inside the chamber, where the latter two may be set to zero for nullifying electrostatic effects. With the test_wallloss module users can confirm that PyCHAM accurately reproduces Fig. 2 of McMurry and Rader (1985), as shown here in Fig. 9, which demonstrates the effect of changing the charge number per particle.

If user sets the McMurry_flag option to zero then a customised particle deposition rate dependence on particle size is available. This option allows application of known or best estimate deposition rates ($\beta$) to the model, as recommended by Wang et al. (2018). Four further inputs are required for this option: the particle diameter (m) at which the inflection in deposition rates occurs ($D_{p,flec}$) (where the inflection point marks a change in dependance of deposition rate with particle size), the rate of particle deposition to wall (s$^{-1}$) at the inflection point ($\beta_{flec}$), and the gradients (s$^{-1}$) of the deposition rate with respect to particle diameter before ($\nabla_{pre}$) and after the inflection ($\nabla_{pro}$), where a linear dependence in log-log space is assumed, consistent with observations (Charan et al., 2019). The equations for deposition rate in this instance are given in Eq. 16, and example dependencies of rate with particle size provided by Fig. 9.

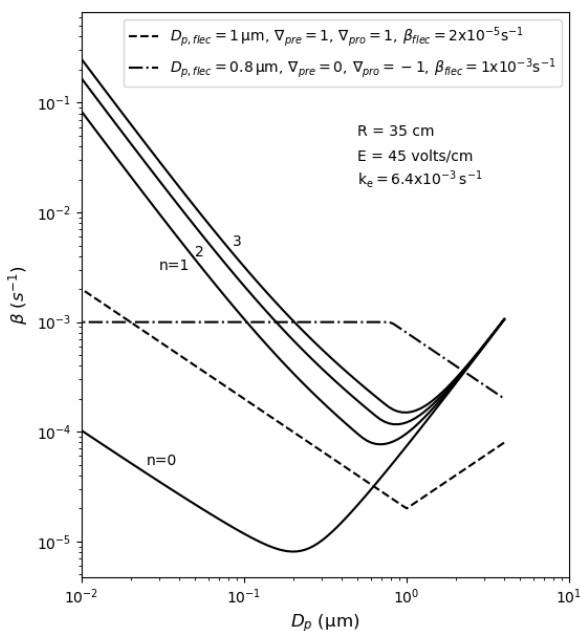

**Figure 9.** Example dependencies of the particle deposition to wall rate using the model of McMurry and Rader (1985) in the solid lines, where the charge per particle is given by n and other inputs given by inset text (R is spherical-equivalent chamber radius, E is the average electric field in the chamber and $k_e$ is the coefficient of eddy diffusion). The dashed lines demonstrate the observation-based deposition rate utility of PyCHAM given in Eq. 16, with inputs at the top of the plot.

$D_p < D_{p,flec}$

$$\log_{10}(\beta(D_p)) = \log_{10}(D_{p,flec}) - \log_{10}(D_p)\nabla_{pre} + \beta_{flec}$$

$D_p \geq D_{p,flec}$

$$\log_{10}(\beta(D_p)) = \log_{10}(D_p) - \log_{10}(D_{p,flec})\nabla_{pro} + \beta_{flec} \tag{16}$$

## 10   Nucleation

The simulation of nucleation to produce newly-formed suspended particles is one of the most active areas of ongoing atmospheric research and many important advances in observing the nucleation process have been, and will continue to be, made through appropriate measurements in chamber experiments and their interpretation (Dada et al., 2020). PyCHAM is not intended to interpret and examine chamber experiments designed to resolve the mechanisms involved in molecular clustering,

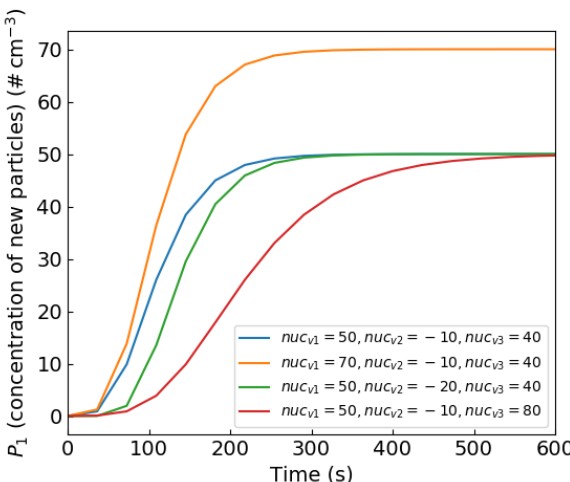

**Figure 10.** Effect of varying the nucleation parameters on simulated particle number concentration when considering only nucleation

nucleation and early growth in particle formation and there are tools much better suited to these processes. However, the use of PyCHAM in simulating chamber processes in the presence of new particle formation necessitates a phenomenological accommodation of the process. Users are therefore able to provide parameters to a Gompertz function for cumulative new particle number, allowing them to fit to observed number size distributions without inferring mechanistic insight:

$$P_1(t) = \mathrm{nuc_{v1}}\left(\exp\left(\mathrm{nuc_{v2}}\left(\exp\left(-t/\mathrm{nuc_{v3}}\right)\right)\right)\right) \tag{17}$$

where $P_1$ ($\#\,\mathrm{cm^{-3}}$) is the number concentration of new particles after time $t$ that enter the smallest size bin, and $\mathrm{nuc_{vn}}$ are the user-defined parameters. As demonstrated in Fig. 10, the resulting function forms an asymmetric sigmoidal curve with time, whilst the parameters allow the amplitude ($\mathrm{nuc_{v1}}$), onset ($\mathrm{nuc_{v2}}$), and duration ($\mathrm{nuc_{v3}}$) of the curve to be adjusted. Fig. 10 is a general example, not related to a specific chemical system as Eq. 17 is independent of chemistry. It is left to the user to vary the nucleation parameters to fit their observations. As Fig. 10 shows, the Gompertz function provides a sigmoidal form beginning with a relatively fast increase in new particle formation before formation rate peaks followed by a relatively slow levelling off. This characteristic is consistent with observations of new particle formation, including Fig. 2B of Riccobono et al. (2014), and other simulations of nucleation, such as Fig. 2B of Dada et al. (2020).

For the moving-centre size structure in PyCHAM, newly nucleated particles determined by Eq. 17 enter the smallest size bin. In contrast, for full-moving, the approach of Roldin et al. (2015) is used where a new, smallest, size bin is formed on every step involving nucleation whilst removing the largest size bin (merging any particles and components present in this bin with those in the second largest size bin).

As with gas-wall partitioning parameters, nucleation parameters should be fitted to measurements by minimising model-observation residuals. For this process the total particle number concentration may be used, however, the greater amount of

data in number size distributions introduces stronger constraint, making it the preferred observation for fitting. In Fig. 11 we demonstrate PyCHAM simulations fitted to observations of a nucleation event from the Manchester aerosol chamber, which had initial concentrations of: $NO_2$ (40 ppb), $O_3$ (60 ppb) and limonene (215 ppb). The experiment was dark at a constant temperature of 298.15 K and relative humidity 50 %. For Eq. 16, parameters were set to: $D_{p,flec} =$1x$10^{-6}$ m, $\beta_{flec} =$6x$10^{-6}$ s$^{-1}$, $\nabla_{pre} =$1 s$^{-1}$ and $\nabla_{pro} =$1 s$^{-1}$, which gave reasonable agreement with observed particle number decay. Here the same chemical scheme as Fig. 1 was used, namely the MCM limonene scheme with appended PRAM scheme.

The smallest size bin for which observations were obtained had a central diameter of 44 nm. Consequently, fitting was performed against measurements for particle sizes greater than those of newly formed particles. The implication for the derived nucleation parameters of Eq. 17 is that if any inaccuracy in non-nucleation processes (gas-particle partitioning, coagulation, particle loss to wall, gas-wall partitioning, gas-phase reaction affecting condensable vapour concentration) is present, these parameters will try to compensate when fitting to observations through minimising observation-simulation residuals. Ideally, therefore, measurements would be available for the concentration of only newly nucleated particles, which would allow the fitted nucleation parameters to be independent of any convoluting process.

To minimise any possible effects from coagulation and particle loss to wall on fitting nucleation parameters, only the first hour of the experiment is considered when estimating residuals. Where necessary, simulation output was linearly interpolated to observation time and particle size points. Observation-simulation residuals ($\sigma$) for the number size distribution ($nsb$) were estimated using:

$$\sigma_{nsb} = \frac{\sum_{t_i=1}^{Z} \sum_{k=1}^{Y} |(n_{lr,t_i,k} - \bar{n}_{t_i,k})|}{\sum_{t_i=1}^{Z} \sum_{k=1}^{Y} (n_{lr,t_i,k} + \bar{n}_{t_i,k})} 100, \tag{18}$$

where $Z$ is the number of time steps, $Y$ is the number of size bins, $t_i$ is the time step index, $k$ is the size bin index and $n_{lr}$ is the particle number concentration from the simulation whilst $\bar{n}$ is that from observations. The denominator is the sum of total particle number concentration from both the simulation and observations. Where number size distributions are in complete disagreement (with results having number concentrations in entirely different size bins), this denominator limits deviation to a helpful (for interpretation) maximum of 100 %. Exact agreement is represented by a $\sigma_{nsb}$ of 0 %.

For a range of Eq. 17 nucleation parameters, Table 1 presents the observation-simulation residuals according to Eq. 18. The temporal profiles of the number size distributions for the entire experiment are shown in Fig. 11, with the model result here from the simulation with minimal residual. In Fig. 11 simulation results are represented by the filled contours, whilst observations are given by the contour lines.

With the shape, size, composition and growth mechanism of the clusters that act as the nucleus of particles subject to ongoing research, in PyCHAM default properties are currently assigned, with a view to advance representation as understanding develops and an appreciation of their physical limitation. An arbitrary involatile component is assumed to form spherical nucleating clusters with a radius of 2 nm. Growth of clusters is assumed to follow gas-particle partitioning (as for particulates of all sizes in PyCHAM). At the current stage of development, this representation of new particle formation in PyCHAM aims to enable simulations of coupled photochemistry and aerosol microphysics in seeded and unseeded experiments. However, a

**Table 1.** Observation-model residuals as defined by Eq. 18 for a dark limonene oxidation experiment without seed particles, with experiment setup described in the main text. Residuals are given for a variety of nucleation parameters, with the minimum residual representing the best fit of simulation to observations.

| $\mathrm{nuc_{v1}}$, $\mathrm{nuc_{v2}}$, $\mathrm{nuc_{v3}}$ | full-moving $\sigma_{nsb}$ (%) | moving-centre $\sigma_{nsb}$ (%) | notes |
|---|---|---|---|
| $2\mathrm{x}10^4$, $-4$, $5\mathrm{x}10^2$ | 74 | 74 | nucleation duration too long $\rightarrow$ reduce $\mathrm{nuc_{v3}}$ |
| $2\mathrm{x}10^4$, $-4\mathrm{x}10^2$, $1\mathrm{x}10^2$ | 63 | 59 | nucleation commences too late $\rightarrow$ reduce $\mathrm{nuc_{v2}}$ |
| $2\mathrm{x}10^4$, $-4$, $1\mathrm{x}10^2$ | 53 | 48 | too few particles newly nucleated $\rightarrow$ increase $\mathrm{nuc_{v1}}$ |
| $3\mathrm{x}10^4$, $-4$, $1\mathrm{x}10^2$ | 40 | 34 | best fit |

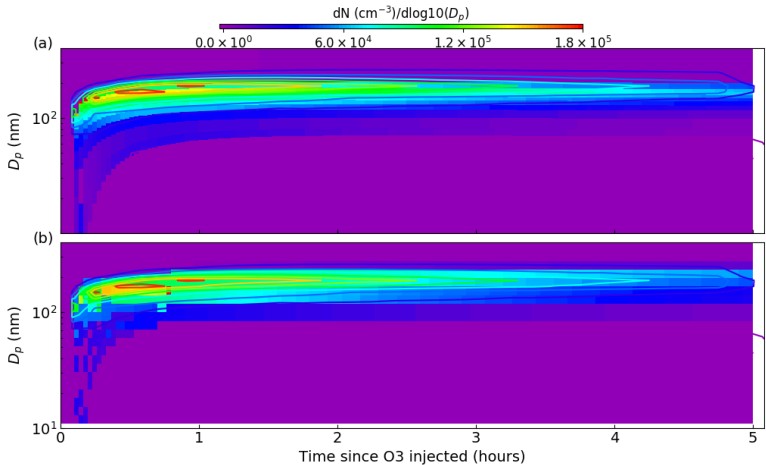

**Figure 11.** Comparing number size distributions from best fit (defined in Table. 1) PyCHAM simulations using the full-moving (a) and moving-centre (b) size structures against observations from a dark limonene oxidation chamber experiment with nucleation. Experiment conditions and simulation setup are given in the main text. Observations are presented as contour lines, whilst model results are filled contours, with both using the colour axis given at the top of the plot. The degree of agreement here is indicated by the difference in these contours at any one point.

more rigorous mechanistic representation of nucleation and early growth should be readily accommodated and will be required before PyCHAM is suitable for investigating new particle formation.

## 11   Sensitivity to temporal resolution and size bin number

In PyCHAM, temporal resolution is set by the user, as represented by the time interval for updating ODE constants, and size bin number is also set by the user (Section 3). Whilst decreasing both resolutions can decrease processing time, inaccuracies may be introduced because PyCHAM processes are sensitive to changes to number size distributions (updated after each time interval) and particle size. Although it is beyond the scope of this paper to assess resolution sensitivity across all possible PyCHAM parameter space, in this section we compare the divergence of outputs from simulations with decreasing temporal resolution and size bin number against a high resolution reference for extremes of the relevant parameter space: seeded experiments with no gas-particle partitioning and both seeded and nucleation experiments with relatively large condensational growth of particles. As in Section 7, two-methylglyceric acid is used in the simulations with partitioning as its vapour pressure at simulation temperature (298 K) makes it semi-volatile. Results here determine the recommended temporal resolution and size bin number, provide a useful illustration of sensitivity and may help users perform sensitivity tests for their individual model inputs.

For the simulations without partitioning, the effect of resolution on particle number size distribution and total number concentration is considered, whilst for the partitioning simulations, concentration of secondary material is also relevant. To allow comparison of low resolution simulations with the high resolution reference, linear interpolation is used to estimate output from the former at the resolution of the latter. Divergence between a low resolution simulation ($lr$) and the high resolution reference is represented by a single absolute percentage deviation ($\sigma$). For number size distribution, deviation ($\sigma_{nsb}$) is found by Eq. 18, with $\bar{n}$ representing the high resolution result.

For total number concentration and total secondary particulate matter concentration, deviation ($\sigma_{lr}$) is calculated as the percentage deviation:

$$\sigma_{lr} = \frac{\sum_{t_i=1}^{Z} |(H_{lr,t_i} - \bar{H}_{t_i})|}{\sum_{t_i=1}^{Z} \vee(H_{lr,t_i}, \bar{H}_{t_i})} 100, \tag{19}$$

where $Z$ is the total number of time steps and $H$ represents either total number concentration or total secondary particulate matter concentration. The use of $\vee(H_{lr,t_i}, \bar{H}_{t_i})$ (the maximum of either the low resolution or high resolution result for a given time step is used for the summation) for the denominator means that when one result is zero and the other is above zero (maximum possible disagreement) a helpful (for interpretation) $\sigma_{lr}$ of 100 % is produced, whilst complete agreement gives a $\sigma_{lr}$ of 0 %.

For simulations assessing sensitivity to temporal resolution, 128 logarithmically spaced size bins are used, for which Fig. 8 indicates no limitation to accuracy due to size bin number. For seeded simulations, we use the same initial number size distribution as in Fig. 8, as this gives a relatively broad range of particle sizes, which is necessary to fully appreciate the size-dependent effects of coagulation, particle loss to wall and nucleation. All simulations were run for 24 hours and the reference simulation had an update time interval of 6 s.

Results for each scenario (seeded with no partitioning; seeded with partitioning; nucleation with partitioning) are provided in Tables 2-4. The first, given in Table 2 represents the no partitioning case, with sensitivity assessed for two setups: only coagulation, and both coagulation and wall loss turned on. Coagulation proceeds as described in Section 8, whilst wall loss is described in Section 9, with the following inputs to recreate a size-dependent wall loss profile similar to n=3 in Fig. 9: $D_{p,flec}$ = 1.0 μm, $\beta_{flec}$ = 1.0x10$^{-4}$ s$^{-1}$, $\nabla_{pre}$ = $\nabla_{pro}$ = 1.5. Consequently, the particle loss to wall is relatively large and the sensitivity results are conservative. Table 2 indicates that under this scenario, particle loss to walls considerably increases sensitivity to temporal resolution compared to the only coagulation case, with much finer resolutions required to achieve average deviations of 10 % or less.

For Table 3, two-methylglyceric acid is introduced at a rate of $1.0$x$10^{-2}$ ppb s$^{-1}$. Comparing the NSD columns in Tables 2 and Table 3, including gas-particle partitioning acts to increase sensitivity to temporal resolution. However, there is less change overall for sensitivity of $N$ and [SPM]. A resolution of around 180 s is required to attain deviations around 10 % or less for total number concentration and secondary material, whilst for number size distribution, not even the lowest temporal resolution of 20 s can agree within 10 % of the reference case of 6 s. This reflects the steep gradients of particle number with particle size that are generated during intense condensational growth periods (e.g. Fig. 6). This effect is also evident when an extremely low volatility organic component is injected at the simulation start at 1 ppb to act as a nucleating agent in an unseeded simulation, with results given in Table 4. We use the nucleation parameters for Eq. 17 of: $\mathrm{nuc_{v1}}$ = 1x10$^4$, $\mathrm{nuc_{v2}}$ = $-1$x10$^1$ and $\mathrm{nuc_{v3}}$ = 1x10$^2$, for a relatively rapid nucleation period (lasting only ten minutes) and therefore conservative assessment of sensitivity. Table 4 shows that, the total number and secondary material concentrations deviate from the reference case by around 10 % for an update time interval of between 20 and 180 s. Given the conservative nature of these simulations we therefore recommend a maximum update time interval of 60 s. However, with the coagulation case effectively representing zero wall loss and showing considerably less deviation across all scenarios, if users can demonstrate relatively low particle wall loss, a coarser resolution could be applied.

In the bottom section of Tables 2-4 are presented the sensitivities to size bin number using the same simulation scenarios as applied to temporal resolution sensitivity. Using a fixed temporal resolution of 60 s, results show the divergence of 8, 32 and 64 size bins compared against results for 128 size bins (all logarithmically spaced). In Table 2, when coagulation alone is effective, reasonable agreement is seen for both factors (NSD and $N$) across all numbers of size bins, with a slight increase in sensitivity when wall loss is also considered. Results for the partitioning cases in Tables 3-4 show that both with and without wall loss, total number concentration and secondary particulate matter concentration give reasonable agreement of around 10 % or less when 8 size bins are used. The exception nucleation with wall loss considered where the large deviation seen for 8 size bins is due to a relatively rapid loss of particles in the 8 size bin run. Deviation of NSD is relatively high for the partitioning cases, even with 64 size bins, therefore 128 size bins must be used when number size distribution is important. In other cases, whilst recognising substantial differences between scenarios and user requirements, we recommend a size bin number of 32 when substantial deviation in the number size distribution is acceptable.

Whilst the resolution sensitivity tests above used a 1 reaction chemical scheme (ensuring that two-methylglyceric acid is recognised), Table 5 demonstrates processing times using the $\alpha-$pinene ozonolysis scheme of the MCM, which comprises

**Table 2.** % deviation from the high resolution reference case for the seed without partitioning (non-volatile particles in the absence of vapours) scenario for number size distribution (NSD) (Eq. 18) and total particle number concentration ($N$) (Eq. 19). In the top section of the table is sensitivity to temporal resolution (update time interval) using 128 size bins, whilst the bottom section is sensitivity to size bin number (# size bins) using a 60 s update time interval. Columns headed coag. are for simulations where coagulation acts alone whilst columns headed coag. & wall are from simulations with both coagulation and particle loss to walls. To aid interpretation deviations less than 10 % are coloured yellow.

| update time interval (s) | NSD, coag. | $N$, coag. | NSD, coag. & wall | $N$, coag. & wall |
|---|---|---|---|---|
| 20 | 0 | 0 | 1 | 1 |
| 180 | 0 | 0 | 21 | 28 |
| 1800 | 2 | 2 | 83 | 49 |
| 43200 | 29 | 31 | 99 | 64 |
| # size bins | NSD, coag. | $N$, coag. | NSD, coag. & wall | $N$, coag. & wall |
| 8 | 14 | 3 | 25 | 13 |
| 32 | 6 | 0 | 8 | 4 |
| 64 | 5 | 5 | 5 | 2 |

**Table 3.** % deviation from the high resolution reference case for the seed with partitioning scenario. Condensation occurs due to the continuous injection of two-methylglyceric acid vapour at a rate of $1.0\text{x}10^{-2}\,\mathrm{ppb\,s^{-1}}$. In addition to the particle properties described in Table 2, the concentration of secondary particulate matter ([SPM]) is also evaluated here using Eq. 19. To aid interpretation deviations less than 10 % are coloured yellow.

| update time interval (s) | NSD, coag. | $N$, coag. | [SPM], coag. | NSD, coag. & wall | $N$, coag. & wall | [SPM], coag. & wall |
|---|---|---|---|---|---|---|
| 20 | 32 | 3 | 2 | 21 | 1 | 4 |
| 180 | 29 | 0 | 2 | 29 | 12 | 8 |
| 1800 | 33 | 1 | 2 | 80 | 51 | 43 |
| 43200 | 77 | 10 | 30 | 99 | 50 | 50 |
| # size bins | NSD, coag. | $N$, coag. | [SPM], coag. | NSD, coag. & wall | $N$, coag. & wall | [SPM], coag. & wall |
| 8 | 72 | 11 | 0 | 68 | 12 | 5 |
| 32 | 53 | 3 | 0 | 49 | 2 | 1 |
| 64 | 48 | 9 | 0 | 43 | 2 | 1 |

approximately $1\text{x}10^3$ reactions. Relevant combinations of temporal resolution and size bin number are provided for a 6 hour unseeded experiment with $\alpha-\mathrm{pinene}$ and ozone introduced at the start to generate a nucleation episode. A 2.5 GHz Intel Core

**Table 4.** % deviation from the high resolution reference case for the nucleation with partitioning scenario. The particle properties given in column headings are defined in Table 3. An extremely low volatility organic component was present at simulation start at 1 ppb, and set as the nucleating component in the absence of seed particles. In addition, two-methylglyceric acid vapour was injected at a rate of $1.0\text{x}10^{-2}\,\text{ppb s}^{-1}$ to grow particles through condensation. To aid interpretation deviations less than 10 % are coloured yellow.

| update time interval (s) | NSD, coag. | $N$, coag. | [SPM], coag. | NSD, coag. & wall | $N$, coag. & wall | [SPM], coag. & wall |
|---|---|---|---|---|---|---|
| 20 | 18 | 1 | 3 | 36 | 8 | 4 |
| 180 | 57 | 8 | 15 | 74 | 15 | 17 |
| 1800 | 88 | 21 | 29 | 99 | 50 | 65 |
| 43200 | 100 | 100 | 100 | 100 | 100 | 100 |
| # size bins | NSD, coag. | $N$, coag. | [SPM], coag. | NSD, coag. & wall | $N$, coag. & wall | [SPM], coag. & wall |
| 8 | 84 | 9 | 0 | 100 | 96 | 100 |
| 32 | 65 | 0 | 0 | 70 | 7 | 2 |
| 64 | 52 | 0 | 0 | 54 | 2 | 1 |

**Table 5.** Processing times (hours, to the nearest tenth) for a 6 hour experiment of $\alpha-$pinene ozonolysis including nucleation. Size bin numbers are in columns and temporal resolutions are in rows.

| Update time interval (s)/# size bins | 2 | 8 | 32 |
|---|---|---|---|
| $6\text{x}10^{1}$ | 0.7 | 2.8 | 44.0 |
| $6\text{x}10^{2}$ | 0.1 | 0.6 | 13.9 |
| $6\text{x}10^{3}$ | 0.0 | 0.1 | 1.8 |

i5 processor was used. It is appreciated that the relatively large processing times in the final column of Table 5 are prohibitively high and could reduce the utility of PyCHAM. This is an important result because it indicates the current limits of employing an interpreted (rather than compiled, e.g. Fortran) programming language. To emphasise this point the processing time for AtChem2, which is a Fortran programme, to produce Fig. 3, namely a gas phase only simulation for 12 hours of $\alpha-$pinene ozonolysis, is 19 s, whereas PyCHAM takes 348 s - a factor of 18 longer. This is with the same tolerances provided to the ODE solver of each model and a 60 s update time interval used for each.

Whilst a compiled language would reduce processing time it would inhibit portability, which we have not compromised with PyCHAM. The slowest section of PyCHAM to process is solution of the ODEs; here the code is already vectorised to optimise speed. A just in time compiler, offers a portable solution to python acceleration, and the core development team are investigating the feasibility of incorporating this for future versions.

## 12 Conclusions

The PyCHAM (CHemistry with Aerosol Microphysics in Python) software for aerosol chambers has been described. Its open source repository is given in Section 2. PyCHAM has been designed for optimal ease of use (from online access to output) whilst being broadly able to address scientific problems of current relevance across a range of aerosol chamber and experimental configurations (Section 2). We have provided a model output for the dark oxidation of limonene to illustrate the coupling of modelled processes: gas-phase chemistry, gas-particle partitioning, gas-wall partitioning, redistribution of particles following change in size, particle loss to wall, coagulation and nucleation (Sections 2 and 3).

The steps to run a simulation using the software's GUI were described in Section 3 and the methods for estimating or setting component properties explained in Section 4. The setting up and solution of gas-phase photochemical reactions is detailed in Section 5, including comparison against the AtChem2 model (Sommariva et al., 2018) for verification and illustration of the effect of varying temporal resolution on model output for a system subject to varying natural light intensity.

In Section 6, gas-particle partitioning along with the moving-centre and full-moving size structures for dealing with changing number size distributios was introduced and assessed against benchmark simulations. For gas-wall partitioning this paper details (Section 7) a parameterisation that aims to satisfy the breadth of chamber characteristics and recommends a method for tuning to observations.

Coagulation was detailed in Section 8 and shown to introduce negligible loss of mass for a relatively complex initial number size distribution after 6 hours despite using a non-mass conserving equation. However, the development team intend to provide an option for a mass conserving treatment in future PyCHAM versions. With Section 9, the three options for treating particle losses to walls were detailed and the resulting deposition rates as a function of particle diameter were exemplified, including assessment against the benchmark of McMurry and Rader (1985). Similar to gas-wall partitioning, nucleation in PyCHAM is treated with a parameterisation that aims to optimise model versatility, with examples of parameter effects provided (Section 10).

In Section 11 the sensitivity of key outputs to temporal resolution and size bin number were illustrated and informed our recommendations of a minimum update time interval of 60 s and a minimum of 32 size bins. We also show a high sensitivity of model accuracy to the rate of particle wall loss and note that users could use lower resolutions if wall loss is lower than used in our tests. This section illustrates that at high resolutions processing time with PyCHAM can decrease its utility, therefore we suggest future work to investigate the use of just in time compilers.

Papers in preparation demonstrate further the utility of PyCHAM and its evaluation when assessed against observations. These papers include both phenomenological and mechanistic approaches to coupled photochemistry and aerosol microphysics, both of which the model readily accommodates.

*Code availability.* The PyCHAM software, figures in this manuscript and code to plot figures is available at: https://github.com/simonom/PyCHAM.

## Appendix A: Model variable inputs

Below is the table of model variables required for input to PyCHAM accompanied by a description.

| Input Name | Description |
| --- | --- |
| res_file_name | Name of folder to save results to |
| total_model_time | Total experiment time to be simulated (s) |
| update_step | Time interval (s) for updating ordinary differential equation constants. Default is 60 s. Can be set to more than the total_model_time variable above to allow uninterrupted integration. |
| recording_time_step | Time interval (s) for recording results. Default is 60 s. |
| size_structure | Determines whether to use the moving-centre (0) or full-moving size structure (1), defaults to moving-centre. |
| number_size_bins | Number of size bins (excluding wall); to turn off particle considerations set to 0 (which is also the default), likewise set pconc and seed_name variables below off. Must be integer (e.g. 1) not float (e.g. 1.0). |
| lower_part_size | Radius of smallest size bin boundary (um) |
| upper_part_size | Radius of largest size bin boundary (um) |
| space_mode | Set to lin for linear spacing of size bins in radius space, or to log for logarithmic spacing of size bins in radius space, if empty defaults to linear spacing |
| wall_on | Determines whether gas-wall partitioning and particle deposition on (1) or off (0), defaults to on. |
| mass_trans_coeff | Mass transfer coefficient of vapour-wall partitioning (/s), if left empty defaults to zero |
| eff_abs_wall_massC | Effective absorbing wall mass concentration (g/m3 (air)), if left empty defaults to zero |
| temperature | Air temperature inside the chamber (K). At least one value must be given for the experiment start (times corresponding to temperatures given in tempt variable below). If multiple values, representing temperatures at different times, then separate with a comma. For example, if the temperature at experiment start is 290.0 K and this increases to 300.0 K after 3600.0 s of the experiment, input is 290.0, 300.0. |
| tempt | Times since start of experiment (s) at which the temperature(s) set by the temperature variable above, are reached. Defaults to 0.0 if left empty as at least the temperature at experiment start needs to be known. If multiple values, representing temperatures at different times, then separate with a comma. For example, if the temperature at experiment start is 290.0 K and this increases to 300.0 K after 3600.0 s of the experiment, input is 0.0, 3600.0. |
| continued on next page | |

| Input Name | Description |
| --- | --- |
| p_init | Pressure of air inside the chamber (Pa) |
| rh | Relative Humidity (fraction, 0-1) |
| lat | Latitude (degrees) for natural light intensity (if applicable, leave empty if not (if experiment is dark set light_status below to 0 for all times)) |
| lon | Longitude (degrees) for natural light intensity (if applicable, leave empty if not (if experiment is dark set light_status below to 0 for all times)) |
| DayOfYear | Day of the year for natural light intensity (if applicable, leave empty if not (if experiment is dark set light_status below to 0 for all times)), must be integer between 1 and 365 |
| daytime_start | Time of the day (s since midnight) for natural light intensity (if applicable, leave empty if not (if experiment is dark set light_status below to 0 for all times)) |
| act_flux_file | Name of csv file stored in PyCHAM/photofiles containing actinic flux values; use only if artificial lights inside chamber are used during experiment. The file should have a line for each wavelength, with the first number in each line representing the wavelength in nm, and the second number separated from the first by a comma stating the flux (Photons/cm2/nm/s) at that wavelength. No headers should be present in this file. Example of file given by /PyCHAM/photofiles/Example_act_flux and example of the act_flux_path variable is: act_flux_path = Example_act_flux.csv. Note, please include the .csv in the variable name if this is part of the file name. Defaults to empty. |
| photo_par_file | Name of txt file stored in PyCHAM/photofiles containing the wavelength-dependent absorption cross-sections and quantum yields for photochemistry. If left empty defaults to MCMv3.2, and is only used if act_flux_path variable above is stated. File must be of .txt format with the formatting: J_n_axs wv_m, axs_m J_n_qy wv_M, qy_m J_end where n is the photochemical reaction number, axs represents the absorption cross-section (cm2/molecule), wv is wavelength (nm), _m is the wavelength number, and qy represents quantum yield (fraction). J_end marks the end of the photolysis file. An example is provided in PyCHAM/photofiles/example_inputs.txt. Note, please include the .txt in the file name. |
| continued on next page | |

705

| Input Name | Description |
| --- | --- |
| ChamSA | Chamber surface area (m2), used if the Rader and McMurry wall loss of particles option (Rader_flag) is set to 1 below |
| coag_on | Set to 1 (the default if left empty) for coagulation to be modelled, or set to zero to omit coagulation |
| nucv1 | Nucleation parameterisation value 1 |
| nucv2 | Nucleation parameterisation value 2 |
| nucv3 | Nucleation parameterisation value 3 |
| nuc_comp | Name of component contributing to nucleation (only one allowed), must correspond to a name in the chemical scheme file. Defaults to empty. If empty, the nucleation module (nuc.py) will not be called. |
| new_partr | Radius of newly nucleated particles (cm), if empty defaults to 2.0e-7 cm. |
| inflectDp | The particle diameter (m) at the inflection point of the size-dependent wall deposition rate. |
| Grad_pre_inflect | Negative log10 of the gradient of particle wall deposition rate against the log10 of particle diameter before inflection (/s). For example, for the rate to decrease by an order of magnitude every order of magnitude increase in particle diameter, set to 1. |
| Grad_post_inflect | Log10 of the gradient of particle wall deposition rate against the log10 of particle diameter after inflection (/s). For example, for the rate to increase by an order of magnitude for every order of magnitude increase in particle diameter, set to 1. |
| Rate_at_inflect | Particle deposition rate to wall at the inflection point for size-dependent particle loss to walls (/s) |
| part_charge_num | Average number of charges per particle, only required if the McMurry and Rader (1985) model for particle deposition to walls is selected |
| elec_field | Average electric field inside the chamber (g.m/A.s3), only required if the McMurry and Rader (1985) model for particle deposition to walls is selected |
| McMurry_flag | Set to 0 to use the particle wall loss parameter values given above or 1 to use the McMurry and Rader (1985, doi: 10.1080/02786828508959054) method for particle wall loss, which uses the chamber surface area given by ChamSA above, average number of charges per particle (part_charge_num above) and average electric field inside chamber (elec_field above), defaults to no particle wall loss if empty, similarly -1 turns off particle wall loss |

| Input Name | Description |
| --- | --- |
| C0 | Initial concentrations of any trace gases input at the experiment start (ppb), must correspond to component names in Comp0 variable below. Separate concentrations of multiple components with a comma. |
| Comp0 | Names of trace gases present at experiment start (in the order corresponding to their concentrations in C0). Note, this is case sensitive, with the case matching that in the chemical scheme file. Separate multiple component names with a comma. |
| Ct | Concentrations of component achieved when injected at some time after experiment start (ppb), if multiple values (representing injection at multiple times), please separate with commas. If multiple components are injected after the start time, then this input should comprise the injected concentrations of components with times separated by commas and components separated by semicolons. E.g., if k ppb of component A injected after m seconds and j ppb of component B injected after n (n>m) seconds, then Ct should be k,0;0,j. The value here is the increase in concentration from the moment before the injection to the moment after (ppb) |
| Compt | Name of component injected at some time after experiment start. Note, this is case sensitive, with the case matching that in the chemical scheme file. If more than one component, separate with a comma. |
| injectt | Time(s) at which injections occur (seconds), which corresponds to the concentrations in Ct, if multiple values (representing injection at multiple times), please separate with commas. If multiple components are injected after the start time, then this input should still consist of just one series of times as these will apply to all components. E.g., if k ppb of component A injected after m seconds and j ppb of component B injected after n (n>m) seconds, then this input should be m, n. |
| const_comp | Name of component with continuous gas-phase concentration inside chamber. Note, this is case sensitive, with the case matching that in the chemical scheme file. Defaults to nothing if left empty. To specifically account for constant influx, see const_infl variable below. |
| const_infl | Name of component(s) with continuous gas-phase influx to chamber. Note, this is case sensitive, with the case matching that in the chemical scheme file. Defaults to nothing if left empty. For constant gas-phase concentration see const_comp variable above. Should be one dimensional array covering all components. For example, if component A has constant influx of K ppb/s from 0 s to 10 s and component B has constant influx of J ppb/s from 5 s to 20 s, the input is: const_infl = A, B. |
|  | |

| Input Name | Description |
|---|---|
| const_infl_t | Times during which constant influx of each component given in the const_infl variable occurs, with the rate of their influx given in the Cinfl variable. Should be one dimensional array covering all components. For example, if component A has constant influx of K ppb/s from 0 s to 10 s and component B has constant influx of J ppb/s from 5 s to 20 s, the input is: const_infl_t = 0, 5, 10, 20. |
| Cinfl | Rate of gas-phase influx of components with constant influx (stated in the const_infl variable above). In units of ppb/s. Defaults to zero if left empty. If multiple components affected, their influx rate should be separated by a semicolon, with a rate given for all times presented in const_infl_t (even if this is constant from the previous time step for a given component). For example, if component A has constant influx of K ppb/s from 0 s to 10 s and component B has constant influx of J ppb/s from 5 s to 20 s, the input is: Cinfl = K, K, 0, 0; 0, J, J, 0. |
| dens_Comp | Chemical scheme names of components with a specified density. Separate names with a comma. Default is no components specified here. |
| dens | Specified densities of components (g/cc). Separate densities with a comma. Default is to estimate density based on the Girolami method contained in UManSysProp. |
| vol_Comp | Names of components with vapour pressures to be manually assigned in the volP variable below, names must correspond to those in the chemical scheme file and if more than one, separated by commas. Can be left empty, which is the default. |
| volP | Vapour pressures (Pa) of components with names given in vol_Comp variable above, where one vapour pressure must be stated for each component named in vol_Comp and multiple values should be separated by a comma. Acceptable for inputs to use e for standard notation, such as 1.0e-2 for 0.01 Pa |
| act_comp | Names of components (corresponding to those the chemical scheme file) with activity coefficients stated in act_user variable below (if multiple names, separate with a comma). Must have same length as act_user. |
| act_user | Activity coefficients of components with names given in act_comp variable above, if multiple values then separate with a comma. Must have same length as act_comp. |
| accom_coeff_comp | Names of components (corresponding to names in chemical scheme file) with accommodation coefficients set by the user in the accom_coeff_user variable below, therefore length must equal that of accom_coeff_user. Multiple names must be separated by a comma. For any components not mentioned in accom_coeff_comp, accommodation coefficient defaults to 1.0 |
| continued on next page | |

| Input Name | Description |
|---|---|
| accom_coeff_user | Accommodation coefficients (dimensionless) of the components with names given in the accom_coeff_comp variable above, therefore number of accommodation coefficients must equal number of names, with multiple coefficients separated by a comma. Can be a function of radius (m), in which case use the variable name radius, e.g: for NO2 and N2O5 with accommodation coefficients set to 1.0 and 6.09e-08/Rp, respectively, where Rp is radius of particle at a given time (m), the inputs are: accom_coeff_comp = NO2, N2O5 accom_coeff_user = 1.0, 6.09e-08/radius. For any components not mentioned in accom_coeff_comp, accommodation coefficient defaults to 1.0. |
| partit_cutoff | The product of saturation vapour pressure (Pa) and activity coefficient above which gas-particle partitioning assumed zero. Defaults to empty list so that all components allowed to partition. |
| pconct | Times (seconds) at which seed particles of number concentration given in pconc variable below are introduced to the chamber. If introduced at multiple times, separate times by a semicolon. For example, for a two size bin simulation with 10 and 5 particles/cc in the first and second size bin respectively introduced at time 0 s, and later at time 120 s seed particles of concentration 6 and 0 particles/cc in the first and second size bin respectively are introduced, input is: pconct = 0; 120 (and the number_size_bins variable above = 2). |
| pconc | Either total particle concentration, in which case should be a scalar, or particle concentration per size bin, in which case length should equal number of particle size bins (# particles/cc (air)). If an array of numbers, then separate numbers by a comma. If a scalar, the particles will be spread across size bins based on the values in the std and mean_rad variables below. To turn off particle considerations leave empty. If seed aerosol introduced at multiple times during the simulation, separate times using a semicolon. For example, for a two size bin simulation with 10 and 5 particles/cc in the first and second size bin respectively introduced at time 0 s, and later at time 120 s seed particles of concentration 6 and 0 particles/cc in the first and second size bin respectively are introduced, the input is: pconc = 10, 5; 6, 0 (and the number_size_bins variable above = 2). |
|  | |

| Input Name | Description |
| --- | --- |
| seed_name | Name of component comprising the seed particles, can either be core for a component not present in the chemical scheme file, a name from this file, or H2O for water, note no quotation marks needed |
| seed_mw | Molecular weight of seed component (g/mol), if empty defaults to that of ammonium sulphate - 132.14 g/mol |
| seed_dens | Density of seed material (g/cc), defaults to 1.0 g/cc if left empty |
| seedVr | Volume ratio of component(s) in seed particles, must match length of seed_name, with the ratio of different components separated by a comma. The same ratio is applied to all size bins for all seed injections. Defaults to equal volume contributions from each component given in seed_name. E.g. for two components with ratio 1:4, input would be 1,4 (or equivalent (e.g. 25, 100)). |
| mean_rad | Mean radius of particles (um), defaults to a flag that tells software to estimate mean radius from the particle size bin radius bounds given by lower_part_size and upper_part_size variables above. If more than one size bin the default is the mid-point of each. If the lognormal size distribution is being found (using the std input below), mean_rad should be a scalar representing the mean radius of the lognormal size distribution. If seed particles are introduced at more than one time, then mean_rad for the different times should be separated by a semicolon. For example, if seed particle with a mean_rad of 1.0e-2 um introduced at start and with mean_rad of 1.0e-1 um introduced after 120 s, the input is: mean_rad = 1.0e-2; 1.0e-1 and the pconct input is pconct = 0; 120. |
| std | Geometric mean standard deviation of seed particle number concentration (dimensionless) when scalar provided in pconc variable above, role explained online in scipy.stats.lognorm page, under pdf method: https://docs.scipy.org/doc/scipy/reference/generated/scipy.stats.lognorm.html. If left empty defaults to 1.1. If seed particles introduced after the experiment start, then separate std for different times using a semicolon. For example, if seed particle with a standard deviation of 1.2 introduced at start and with standard deviation of 1.3 introduced after 120 s, the std input is: std = 1.2; 1.3 and the pconct input is: pconct = 0; 120 |
| continued on next page | |

710

| Input Name | Description |
|---|---|
| seed_diss | Dissociation constant for seed component (dimensionless); defaults to 1.0. |
| light_time | Times (s) for lighting condition, corresponding to the elements of the light_status variable below, if empty defaults to lights off for whole experiment. Use this variable regardless of whether light is natural or artificial (chamber lamps). For example, for a 4 hour experiment, with lights on for first half and lights off for second, use: light_time = 0.0, 7200.0. If light_time doesn't include the experiment start (0.0 s), default is lights off at experiment start. |
| light_status | Set to 1 for lights on and 0 for lights off, with times given in the light_time variable above, if empty defaults to lights off for whole experiment. Setting to off (0) means that even if variables above that define light intensity are submitted the simulation will be dark. Use this variable for both natural and artificial (chamber lamps) light. The lighting condition for a particular time is recognised when the simulated time meets the time given in light_time. For example, for a 4 hour experiment, with lights on for first half and lights off for second, use: light_status = 1, 0. If status not given for the experiment start (0.0 s), default is lights off at experiment start. |
| tracked_comp | Name of component(s) to track rate of concentration change (molecules/cc.s); must match name given in chemical scheme, and if multiple components given they must be separated by a comma. Can be left empty and then defaults to tracking no components. |
| umansysprop_update | Flag to update the UManSysProp module via internet connection: set to 1 to update and 0 to not update. If empty defaults to no update. In the case of no update, the module PyCHAM checks whether an existing UManSysProp module is available and if not tries to update via the internet. If update requested and either no internet or UManSysProp repository page is down, code stops with an error. |
| chem_scheme_markers | Markers denoting various sections of the user's chemical scheme. If left empty defaults to Kinetic PreProcessor (KPP) formatting. If filled, must have following elements separated with commas: marker for punctuation at start of reaction lines (just the first element), marker for peroxy radical list starting, punctuation between peroxy radical names, prefix to peroxy radical name, string after peroxy radical name, number of lines taken by peroxy radical list (including the line containing the marker for peroxy radical list starting), punctuation at the end of lines for generic rate coefficients. For example, for the MCM FACSIMILE format: chem_scheme_markers = %, RO2, +, , , 20, ; would be used. |
| continued on next page | |

| Input Name | Description |
| --- | --- |
| int_tol | Integration tolerances, with absolute tolerance first followed by relative tolerance, if left empty defaults to the maximum required during testing for stable solution: 1.0e-3 for absolute and 1.0e-4 for relative |
| dil_fac | Volume fraction per second chamber is diluted by, should be just a single number. Defaults to zero if left empty. |

Table A1 containing the PyCHAM variable inputs and their associated descriptions.

*Author contributions.* Gordon McFiggans was principal investigator for the PyCHAM project. Simon O'Meara and Shuxuan Xu equally contributed to writing of the PyCHAM software. David Topping wrote the PyBOX software, Douglas Lowe and Gerard Capes wrote the MANIC software, both MANIC and PyBOX were used as starting points for PyCHAM. Rami Alfarra and Yunqi Shao provided guidance on chamber experiments. Simon O'Meara wrote this manuscript, with edits provided by Gordon McFiggans, Rami Alfarra, Shuxuan Xu and David Topping

*Competing interests.* The authors declare that they have no conflict of interest.

*Disclaimer.* The PyCHAM software is provided under the GNU General Public License v3.0.

*Acknowledgements.* This project has received funding from the European Union's Horizon 2020 research and innovation programme under grant agreement No 730997 which supports the EUROCHAMP2020 research programme. Authors O'Meara and Alfarra received funding support from the Natural Environment Research Council through the National Centre for Atmospheric Science.

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
