# Peer review of "PyCHAM (v2.1.1): a Python box model for simulating aerosol chambers"

_Geoscientific Model Development, 2020_

## Referee Comment (RC1) · Anonymous Referee #1 · 19 Sep 2020

Review of the manuscript "PyCHAM (v1.3.4): a Python box model for simulating aerosol chambers"

**General comment:**
This manuscript describes and assess the open source Chemistry with Aerosol Microphysics in Python (PyCHAM) box model software for aerosol chambers written in Python. The manuscript is generally very well written. I think PyCHAM has the potential to become a user friendly and widely used box model for analysis and interpretation of smog chamber experiments. The main concern I have is the long simulation time presented in Table 1. I consider that this manuscript should be accepted for publication in Geoscientific Model Development after some minor revision and appropriate answers to all comments.

**Specific comments:**
I could not find any clear description of how the gas-particle partitioning of multiple condensable organic and inorganic vapors are treated in PyCHAM. Do you consider the Kelvin effect and Raoult's law for ideal mixtures? Possible I just missed this information. I think you should add a separate section which describe more in detail how the gas-particle partitioning onto different size bins are treated and how you keep track of the chemical composition in each size bin over time.

Page 3, L78-79 and Fig. 1. Did you set the $N_2O_5$ activity coefficient to zero or should it be unity, e.g. assuming ideal solution for $N_2O_5$. What activity coefficients do you use for the dissolved inorganic ions, e.g. in this case $NO_3^-$, $H^+$? Which base neutralize the dissolved $HNO_3$? How did you calculate the particle water content? How do you calculate the $[H^+]$ in the aqueous phase? $[H^+]$ is needed in order to calculate the effective Henry's law coefficient which in turn is needed to derive the condensation and dissolution growth for $HNO_3$ and $N_2O_5$ I presume.

Based on the previous comments, does PyCHAM include any thermodynamics module which iteratively calculates the aqueous phase [H+] and particle water content etc., and if yes, how and when is it called?

P9-10, section 5.1 and Fig. 3.   Considering that AtChem2 and PyCHAM use exactly the same MCM chemical scheme I am surprised that the deviation in the NO concentration is several percentage. Please discussion these results a bit more. What can the reasons be? Which error tolerances (solvers) did you use in PyCHAM and AtChem2?

P14, 307-308. First, does not particle wall losses also directly influence the particle number size concentration? Secondly, would it not be more appropriate to write particle number concentration size distribution or just particle number size distribution?

P14, L319-L322, I agree that the full-moving particle distribution technique is not ideal when consider addition of new particles to the particle size distribution over time, but it is not impossible to use the full-moving approach as long as you add new particle size bins during the course of the simulation see e.g. Sect. 3.3 in Roldin et al., Atmos. Chem. Phys., 15, 10777–10798, 2015. Hence, I would encourage you to add the option for users of PyCHAM to use the full-moving approach.

Section 7, Fig. 7 I was expecting the accumulation mode in Fig 7a to move towards the right because of condensational growth during the course of the simulation. But maybe this is not apparent when only looking at the particle volume size distribution. Can you also add a panel illustrating the particle number size distribution at the start and in the end? I also think that it would have been more interesting and more appropriate, considering the intended use of PyCHAM, to show similar results from an idealized no-seed secondary organic aerosol formation experiment with new particle formation and consecutive growth.

Section 8. I generally do not like numerical methods which is not mass conserving but I can accept the choice of the semiimplicit method of solving the coagulation process since you show that it is almost mass conserving. Still, I wonder what the advantage is of using this method in front of a simple fixed section method which will be both mass and number conserving? Is the semiimplicit approach by Jacobson (2005) resulting in less numerical diffusion problems compared to methods which are both mass and number conserving, but which rely on redistribution of the exact new single particle volumes onto the existing size bins using a fixed section approach (see e.g. Korhonen et al., Atmos. Chem. Phys., 4, 757–771, 2004)? Since, coagulation in contrast to condensation only change the size of a minor fraction of the total particle concentration each time step, numerical diffusion should not be a major concern with this fixed section approach and hence, I would encourage you to consider using such a mass and number conserving approach in future PyCHAM versions.

Section 11, Fig. 11. To me this is the section which is least easy to understand. I think this is because Fig. 11 is not easy to follow. I think you should avoid to use a dark green background and red lines. I would suggest that you if possible replace this figure with one or several tables. You may then consider to mark the cells with less than 10 % deviation from the reference case.

Table 1. Please write out the actual simulation time in hours instead of log10(time). $10^{3.4}$ s $\approx 0.7$ h for 2 size bins with 60 s time step is quite a long time for 6 hours of simulation I would say. I presume that this is mainly the gas-phase chemistry. I would like you to add the simulation time for a simulation without particles, e.g. only gas-phase chemistry and also add the simulation time for AtChem2 (no particles). I guess AtChem2 is written in Fortran or? Too me $10^{5.2}$ s $\approx 44$ h for 32 size bins with 60 s time step is not very user friendly. I guess it must have to do with the ode-solver and the stiffness of the differential equation system when you add many particle size bins. I would like to see some more discussion about these results and how they could be solved in future versions.

**Possible typos:**
P11, L255. I do not understand the formulation ",with factor increases of 7 and 53 for $6 \times 10^2$ and $6 \times 10^1$ s resolutions, respectively."

P12, L286. "Third, $k_w$ was held whilst" should it be Third, $k_w$ was held constant whilst … ?

---

## Referee Comment (RC2) · Anonymous Referee #2 · 20 Sep 2020

O'Meara et al. described in detail about the structure and demos of an open-source python box model for simulating aerosol chambers (PyCHAM). Comparisons were made with the results from other models. Sensitivities of different parameters were also tested. The manuscript is overall well written and the PyCHAM could be very helpful for both experimentalists and modelers. I suggest for publication in GMD after some minor revisions:

1. What is the file format of MCM that PyCHAM is taking, i.e., can it be directly used after downloaded the specific file from the MCM website or some formatting has to be made to the file before imported?

2. Fig. 1: since the size distribution uses $dN/dlog_{10}(D_p)$, the y-axis should be in $log_{10}$ scale instead of linear scale.

[Figure]

3. Is the chamber temperature coupled with the vapor molecule thermodynamic properties (e.g., the vapor pressure, diffusivity, etc.) in the PyCHAM?

4. What is the ODE solver used in the PyCHAM since most of the ODEs are very stiff? Are Jacobian matrices specified or numerically calculated in the code? What are the units used when simulating, e.g., ppm/min or pg/s or not the same for compounds in different phases?

5. L282: 0.5 $\mu$m seed particles are usually too big in a chamber experiment, which can cause significant particle-wall loss. Usually, the seed particles are around 100 nm. If no particle-wall loss is considered in this section, suggest using a smaller size of particles but with the same particle surface area concentration to test.

6. Given different $C_w$ values are used in Sec.6 for sensitivity test, Krechmer et al. (2016, DOI: 10.1021/acs.est.6b00606, Krechmer 2017 cited in the manuscript is not on this topic) have shown the vapor pressure-dependent $C_w$. Since the PyCHAM couples the molecule structure with the UManSysProp module, this character can be easily incorporated.

7. For both $\alpha$-pinene and isoprene SOA experiments, please list the compounds (as well as their corresponding vapor pressures) that are allowed to partition to the particle phase.

8. L390: What are the timescales of coagulation versus condensation growth?

9. Fig. 10: Which experiment does this figure correspond to, $\alpha$-pinene or isoprene?

10. Eq. 11: What is the unit of $P_1$? From the description, it looks like a rate. But Fig. 10 indicates it is a number concentration. If as a concentration, the nucleation rate would be changing over time (reaching the maximum around 150 s by differentiating curves in Fig. 10), how does that happen? Maybe explain this a bit more in this section.

11. Sec. 11: The title "... spatial resolution" is confusing, which is actually "size" resolution. When appearing together with "temporal", one could think it as in a 3D

space. Suggest to change it to an appropriate word.

12. L471: Which method is used to interpolate the data points?

13. Eq. 13: Maybe use a different symbol (or sub-/super-scripts) for $\sigma_g$ since it has been used to evaluate the $Z$ & $Y$.

14. Fig. 11 is a little bit difficult to follow: a). The simulation time in the context means the time consumed by the processor to finish the simulation, but one could think it as the time in the model to simulate. Maybe change to a different name. b). What is the number of size bin for different time interval simulation? c). If possible, contour plots of derivation/consuming time with the x-axis of the time interval and the y-axis of the number of size bin would be appreciated.

15. Table 1: Instead of using $log_{10}$ unit, units of hr or min are more intuitive to the readers.

---

## Referee Comment (RC3) · Anonymous Referee #3 · 3 Oct 2020

This paper describes the PyChAM model for simulating aerosol chambers. In my opinion, the paper is written well on the whole and the model appears to have been constructed and evaluated rigorously. The results presented appear highly reproducible given the open access nature of the model and the careful description of the source of data for all of the model comparisons.

I have some very minor comments to aid understanding:

1. Line 48: text in brackets should be clearer 2. Line 55-56: can the authors expand on how this fitting procedure is carried out? is there enough information for the user to be able to do this? 3. Line 72: define sufficient. 4. Line 74: what do the authors mean by injected again later? 5. Line 79: give value of accommodation coefficient 6. Figure 1: line up the x-axes to aid interpretation. 7. Line 101 doesn't appear to follow line 100??

[Figure]

8. Line 103: I guess photolysis rates are only constant for indoor lights? 9. Line 129: define process tendencies 10. Line 185: define what relatively low means 11. line 286: is the word 'constant' missing? 12. Line 313 - typo - predominantly?

---

## Author Comment (AC1) · 28 Oct 2020

The comment was uploaded in the form of a supplement:
https://gmd.copernicus.org/preprints/gmd-2020-234/gmd-2020-234-AC1-
supplement.zip
* * *

---

## Author Comment (AC2) · 24 Nov 2020

Addendum to Response to referee comments on the manuscript "PyCHAM (v1.3.4): a Python box model for simulating aerosol chambers"

Motivated by the referee comments during the initial review of this manuscript, the authors were pleased to find that two modifications to the PyCHAM programme reduced processing times for by up to a factor of 500. The first modification is to change from the Assimulo BDF solver to the scipy BDF solver, the second is to integrate the problem for gas-particle partitioning of water prior to integration of other processes (gas-phase photochemistry, gas-particle partitioning of non-water component(s) and gas-wall partitioning of all components).

[Figure]

Including this change we have repeated all the simulations presented in the paper and found no change in model results. The significant acceleration in processing time greatly increases utility for users; for example, the time for the 32 size bin 60 s time step simulation presented in Table 5 now takes 4 minutes rather than 44 hours. We therefore request that referees consider the revised manuscript, for which the relevant changes are detailed below.

Change Number Change Detail 1 The model version is updated to 2.1.1 in the manuscript title 2 Line 120: Reference of Assimulo ODE solver replaced by reference to SciPy ODE solver 3 Line 293: processing times for the gas-phase photochemistry problem updated 4 Table 5: processing times updated 5 Line 660: discussion around processing times revised to be consistent with Table 5: "Whilst the processing times in Table 5 are reasonable, it is appreciated that higher resolutions and more complex chemical schemes may be used. Future work will investigate use of a just in time compiler, which offers a portable solution to python acceleration."

6 Line 688: removed comment about long processing times decreasing utility

---

## Author Response (AR1)

Response to referee comments on the manuscript "PyCHAM (v1.3.4): a Python box model for simulating aerosol chambers"

The authors thank the referees for their thoughtful and helpful comments. In response we have improved the manuscript and PyCHAM programme. The changes have resulted in an updated PyCHAM programme, for which we have updated the version number to v2.0.4, which is reflected in the new manuscript title. Another author – Yunqi Shao – has been added to the author list and acknowledgement section for her contribution to the improved nucleation section of the paper.

In the table below we address each referee comment. In the response column references to lines pertain to the revised paper and line numbers are inclusive.

We provide a modification in the revised paper that addresses the natural tendency for readers to compare outputs in this description paper against observations from experiments. Lines 68-71 emphasis the purpose and limits of the presented results: "Comparison of some results presented below with experimental observations is possible, however it is beyond the scope of this paper to investigate the accuracy of chemical mechanisms or estimation methods that PyCHAM can use, rather the examples below illustrate the utility of PyCHAM to test the sensitivity of such techniques".

| Referee Comment | Response |
|---|---|
| Referee # 1 | |
| 1. I could not find any clear description of how the gas-particle partitioning of multiple condensable organic and inorganic vapors are treated in PyCHAM. Do you consider the Kelvin effect and Raoult's law for ideal mixtures? Possible I just missed this information. I think you should add a separate section which describe more in detail how the gas-particle partitioning onto different size bins are treated and how you keep track of the chemical composition in each size bin over time. | This information was very briefly mentioned in the original draft and we agree that further detail is beneficial.

Lines 302-316 now provide an explanation. |
| 2. Page 3, L78-79 and Fig. 1. Did you set the N2O5 activity coefficient to zero or should it be unity,

e.g. assuming ideal solution for N2O5. What activity coefficients do you use for the dissolved

-+
inorganic ions, e.g. in this case NO3 , H ? Which base neutralize the dissolved HNO3? How did | Whilst trying to provide a realistic illustration of PyCHAM application to chamber experiments in a concise fashion, the referees have noted that the detail around Fig. 1 is insufficient.

We have clarified and added information to the corresponding simulation explanation in lines 72-100. This includes explanation of simulation N2O5 hydrolysis and estimating the |

| | |
|---|---|
| you calculate the particle water content? How do you calculate the $[H^+]$ in the aqueous phase? $[H^+]$ is needed in order to calculate the effective Henry's law coefficient which in turn is needed to derive the condensation and dissolution growth for HNO3 and N2O5 I presume.

Based on the previous comments, does PyCHAM include any thermodynamics module which iteratively calculates the aqueous phase [H+] and particle water content etc., and if yes, how and when is it called? | accommodation coefficient for N2O5. It also includes detail on the absence of particle-phase chemistry or thermodynamic property estimation in PyHCAM – an important limitation that is now emphasised by these modifications. |
| 3. P9-10, section 5.1 and Fig. 3. Considering that AtChem2 and PyCHAM use exactly the same MCM chemical scheme I am surprised that the deviation in the NO concentration is several percentage. Please discussion these results a bit more. What can the reasons be? Which error tolerances (solvers) did you use in PyCHAM and AtChem2? | The authors found that the wrong day of year had been used in the PyCHAM simulations for this comparison with AtChem2.

The new results are shown in Figs. 3 & 4, with Fig. 3 showing that deviation reduced to around 2 % with this correction.

Line 269 gives tolerances. |
| 4. P14, 307-308. First, does not particle wall losses also directly influence the particle number size concentration? Secondly, would it not be more appropriate to write particle number concentration size distribution or just particle number size distribution? | Corrections made on line 302 |
| 5. P14, L319-L322, I agree that the full-moving particle distribution technique is not ideal when consider addition of new particles to the particle size distribution over time, but it is not impossible to use the full-moving approach as long as you add new particle size bins during the course of the simulation see e.g. Sect. 3.3 in Roldin et al., Atmos. Chem. Phys., 15, 10777– 10798, 2015. Hence, I would encourage you to add the option for users of PyCHAM to use the full-moving approach. | We thank the referee for this detail and have provided the full-moving size structure as an option in the new version of PyCHAM.

Results involving a test/example of full-moving are presented in Figs. 6 and 11. |

| | |
|---|---|
| 6. Section 7, Fig. 7 I was expecting the accumulation mode in Fig 7a to move towards the right because of condensational growth during the course of the simulation. But maybe this is not apparent when only looking at the particle volume size distribution. Can you also add a panel illustrating the particle number size distribution at the start and in the end? | Fig. 7 has become Fig. 6 to allow explanation of gas-particle partitioning prior to the section on gas-wall partitioning

Plot (a) now contains both the volume- and the number-size distributions |
| 7. I also think that it would have been more interesting and more appropriate, considering the intended use of PyCHAM, to show similar results from an idealized no-seed secondary organic aerosol formation experiment with new particle formation and consecutive growth. | The authors agree and have used a simple nucleation experiment from the Manchester aerosol chamber in the new Fig. 11 and Table 1 to compare PyCHAM against and to illustrate fitting of nucleation parameters.

Whilst the main objective of section 10 is describing and exemplifying nucleation treatment in PyCHAM. Fig. 11 also acts to demonstrate a comparison of simulation output for a new particle formation and consecutive growth experiment with observations. |
| 8. Section 8. I generally do not like numerical methods which is not mass conserving but I can accept the choice of the semiimplicit method of solving the coagulation process since you show that it is almost mass conserving. Still, I wonder what the advantage is of using this method in front of a simple fixed section method which will be both mass and number conserving? Is the semiimplicit approach by Jacobson (2005) resulting in less numerical diffusion problems compared to methods which are both mass and number conserving, but which rely on redistribution of the exact new single particle volumes onto the existing size bins using a fixed section approach (see e.g. Korhonen et al., Atmos. Chem. Phys., 4, 757–771, 2004)? Since, coagulation in contrast to condensation only change the size of a minor fraction of the total particle concentration each | This comment prompted us to detail further the semiimplicit method and the reason for this choice of method. Furthermore, we will investigate the feasibility of a mass-conserving approach. The relevant manuscript lines are 438-469.

We also make the limitation of the semiimplicit approach clearer in the conclusion at line 686. |

| | |
|---|---|
| time step, numerical diffusion should not be a major concern with this fixed section approach and hence, I would encourage you to consider using such a mass and number conserving approach in future PyCHAM versions. | |
| 9. Section 11, Fig. 11. To me this is the section which is least easy to understand. I think this is because Fig. 11 is not easy to follow. I think you should avoid to use a dark green background and red lines. I would suggest that you if possible replace this figure with one or several tables. You may then consider to mark the cells with less than 10 % deviation from the reference case. | Following this suggestion, we have substituted the original Fig. 11 for the new Tables 2-4.

The contours of processing time provided in Fig. 11 were not particularly relevant due to the very simple chemical scheme used, therefore we leave Table 5 as an illustration of relevant processing times. |
| 10. Table 1. Please write out the actual simulation time in hours instead of $\log 10$(time). $10^{3.4}$ s $\approx 0.7$ h for 2 size bins with 60 s time step is quite a long time for 6 hours of simulation I would say. I presume that this is mainly the gas-phase chemistry. I would like you to add the simulation time for a simulation without particles, e.g. only gas-phase chemistry and also add the simulation time for AtChem2 (no particles). I guess AtChem2 is written in Fortran or? Too me $10^{5.2}$ s $\approx 44$ h for 32 size bins with 60 s time step is not very user friendly. I guess it must have to do with the ode-solver and the stiffness of the differential equation system when you add many particle size bins. I would like to see some more discussion about these results and how they could be solved in future versions. | Table 1 is now Table 5. Presentation of processing times has been altered as suggested.

Lines 657-670 make greater emphasis of the processing time limitation in PyCHAM, including comparison with AtChem for a like-for-like simulation.

In these lines and in the conclusion (line 696) we suggest future work investigating incorporation of just in time compilers. |
| 11. P11, L255. I do not understand the formulation ",with factor increases of 7 and 53 for $6x10^2$ and $6x10^1$ s resolutions, respectively." | Corrected in line 295 |

| | |
|---|---|
| 12. P12, L286. "Third, $k_w$ was held whilst" should it be Third, $k_w$ was held constant whilst ... ? | Corrected on line 393 |
| Referee # 2 | |
| 1. What is the file format of MCM that PyCHAM is taking, i.e., can it be directly used after downloaded the specific file from the MCM website or some formatting has to be made to the file before imported? | This explanation is now provided in line 148-151. |
| 2. Fig. 1: since the size distribution uses $dN/dlog_{10}(D_p)$, the y-axis should be in $log_{10}$ scale instead of linear scale. | Fig. 1 now has this improvement |
| 3. Is the chamber temperature coupled with the vapor molecule thermodynamic prop- erties (e.g., the vapor pressure, diffusivity, etc.) in the PyCHAM? | Clarity around the effect of changing temperature is provided in line 183 |
| 4. What is the ODE solver used in the PyCHAM since most of the ODEs are very stiff? Are Jacobian matrices specified or numerically calculated in the code? What are the units used when simulating, e.g., ppm/min or pg/s or not the same for compounds in different phases? | These very important features are now better described in lines 116-121 |
| 5. L282: 0.5 µm seed particles are usually too big in a chamber experiment, which can cause significant particle-wall loss. Usually, the seed particles are around 100 nm. If no particle-wall loss is considered in this section, suggest using a smaller size of particles but with the same particle surface area concentration to test. | The authors agree that where possible simulations should represent those in real chamber experiments. Therefore, we use 100 nm diameter particles, whilst increasing number concentration to maintain surface area. The new results are shown in Fig. 7.

This comment had referred to section 6 in the original draft, but this is now section 7 so that a description of gas-particle partitioning could precede it. In section 7.1 now we provide the updated Fig. 7 as a response and have adapted the main text (lines 385-390) accordingly. |

| 6. Given different $C_W$ values are used in Sec.6 for sensitivity test, Krechmer et al. (2016, DOI: 10.1021/acs.est.6b00606, Krechmer 2017 cited in the manuscript is not on this topic) have shown the vapor pressure-dependent $C_W$. Since the PyCHAM couples the molecule structure with the UManSysProp module, this character can be easily incorporated. | We thank the referee for their insight on this topic and have altered the references accordingly. We cite the mentioned paper but explain in lines 365-368 why its associated parametrisation for $C_w$ is not included in this paper. |
|---|---|
| 7. For both α-pinene and isoprene SOA experiments, please list the compounds (as well as their corresponding vapor pressures) that are allowed to partition to the particle phase. | This comment has prompted a significant upgrade to PyCHAM utility: a plotting script for a volatility basis set analysis of component contribution to particle-phase mass as a function of time and production of a comma separated value file containing individual component names, vapour pressures and their particle-phase mass concentration as a function of time.

Using this plotting tool, we have included plot b in Fig. 7, which gives a time-dependent summary of mass contributions from components lumped by volatility. This plot prompts discussion in lines 399-424 which readers may find helpful in their own simulations.

Whilst section 7 covers the isoprene experiment mentioned by the referee, the alpha-pinene simulations in section 5 deal only with the gas-phase, which we clarify now on line 270.

The simulations using MCM schemes involve hundreds of components, all of which are allowed to partition to particles, which we now clarify on line 310. In the interest of readability we therefore do not provide the list suggested by the referee in the paper, however we hope that the new tool described above, which performs the same function with additional information and prompts new discussion in the paper (as detailed above), satisfies. |

| | |
|---|---|
| 8. L390: What are the timescales of coagulation versus condensation growth? | Line 495 includes this information which improves this section. |
| 9. Fig. 10: Which experiment does this figure correspond to, α-pinene or isoprene? | Line 546 now better explains that this is a general plot independent of chemical scheme |
| 10. Eq. 11: What is the unit of $P_1$? From the description, it looks like a rate. But Fig. 10 indicates it is a number concentration. If as a concentration, the nucleation rate would be changing over time (reaching the maximum around 150 s by differentiating curves in Fig. 10), how does that happen? Maybe explain this a bit more in this section. | The unit is now provided in line 544, whilst we better discuss the implications of Eq. 17 for rate of new particle formation in lines 548-550. |
| 11. Sec. 11: The title "... spatial resolution" is confusing, which is actually "size" resolution. When appearing together with "temporal", one could think it as in a 3D space. Suggest to change it to an appropriate word. | The authors agree and have changed to number of size bins. |
| 12. L471: Which method is used to interpolate the data points? | Linear interpolation is now specified on line 607 |
| 13. Eq. 13: Maybe use a different symbol (or sub-/super-scripts) for $\sigma_g$ since it has been used to evaluate the Z & Y . | A new subscript has been used in Eq. 18 to make the labelling more intuitive. |
| 14. Fig. 11 is a little bit difficult to follow: a). The simulation time in the context means the time consumed by the processor to finish the simulation, but one could think it as the time in the model to simulate. Maybe change to a different name. b). What is the number of size bin for different time interval simulation? c). If possible, contour plots of derivation/consuming time with the x-axis of the time interval and the y-axis of the number of size bin would be appreciated. | The Fig. 11 of the original draft has been transformed to Tables 2-4, following both this comment and that of referee # 1.

Simulation time has been changed to processing time, which is defined at line 134

The processing times provided in Fig. 11 of the original draft were not particularly helpful as they related to a chemical scheme with only one |

| | equation. Instead Table 5 has been improved and its surrounding discussion around processing times improved (lines 657-670). |
|---|---|
| 15. Table 1: Instead of using $\log_{10}$ unit, units of hr or min are more intuitive to the readers. | Change made, note the original table 1 is now table 5. |
| Referee # 3 | |
| 1. Line 48: text in brackets should be clearer | Line 49 is now clearer and more accurate, since nucleation is allowed to occur in both seeded and unseeded experiments |
| 2. Line 55-56: can the authors expand on how this fitting procedure is carried out? is there enough information for the user to be able to do this? | Lines 56-58 clarify processes requiring fitting and detail where further information can be found. |
| 3. Line 72: define sufficient. | The entire introduction to Fig. 1 (lines 72-81) has been improved and the comment on sufficient concentrations for nucleation was seen as unnecessary, therefore please see the rewording on line 79. |
| 4. Line 74: what do the authors mean by injected again later? | Clarity improved on line 80. |
| 5. Line 79: give value of accommodation coefficient | Eq. 1 is provided and its surrounding text (lines 88-95) now provides more information about the accommodation coefficient for dinitrogen pentoxide |
| 6. Figure 1: line up the x-axes to aid interpretation. | We thank the referee for this improvement which is provided in the revised Fig. 1 |
| 7. Line 101 doesn't appear to follow line 100?? | Lines 123-126 provide improved wording of this line to aid readability. |
| 8. Line 103: I guess photolysis rates are only constant for indoor lights? | Clarified on line 124. |
| 9. Line 129: define process tendencies | Lines 159-160 now include a definition. |
| 10. Line 185: define what relatively low means | Line 222 now includes a definition |

| | |
|---|---|
| 11. line 286: is the word 'constant' missing? | Corrected on line 393 |
| 12. Line 313 - typo - predominantly? | Corrected on line 317. |

model version number updated following updates recommended by referees

[revised manuscript text omitted]
} = 1\text{x}10^{-6}\,\text{m}$, $\beta_{flec} = 6\text{x}10^{-6}\,\text{s}^{-1}$, $\nabla_{pre} = 1\,\text{s}^{-1}$ and $\nabla_{pro} = 1\,\text{s}^{-1}$, which gave reasonable agreement with observed particle number decay. Here the same chemical scheme as Fig. 1 was used, namely the MCM limonene scheme with appended PRAM scheme.

The smallest size bin for which observations were obtained had a central diameter of 44 nm. Consequently, fitting was performed against measurements for particle sizes greater than those of newly formed particles. The implication for the derived nucleation parameters of Eq. 17 is that if any inaccuracy in non-nucleation processes (gas-particle partitioning, coagulation, particle loss to wall, gas-wall partitioning, gas-phase reaction affecting condensable vapour concentration) is present, these parameters will try to compensate when fitting to observations through minimising observation-simulation residuals. Ideally, therefore, measurements would be available for the concentration of only newly nucleated particles, which would allow the fitted nucleation parameters to be independent of any convoluting process.

To minimise any possible effects from coagulation and particle loss to wall on fitting nucleation parameters, only the first hour of the experiment is considered when estimating residuals. Where necessary, simulation output was linearly interpolated to observation time and particle size points. Observation-simulation residuals ($\sigma$) for the number size distribution ($nsb$) were estimated using:

$$\sigma_{nsb} = \frac{\sum_{t_i=1}^{Z} \sum_{k=1}^{Y} |(n_{lr,t_i,k} - \bar{n}_{t_i,k})|}{\sum_{t_i=1}^{Z} \sum_{k=1}^{Y} (n_{lr,t_i,k} + \bar{n}_{t_i,k})} 100, \tag{18}$$

where $Z$ is the number of time steps, $Y$ is the number of size bins, $t_i$ is the time step index, $k$ is the size bin index and $n_{lr}$ is the particle number concentration from the simulation whilst $\bar{n}$ is that from observations. The denominator is the sum of total particle number concentration from both the simulation and observations. Where number size distributions are in complete disagreement (with results having number concentrations in entirely different size bins), this denominator limits deviation to a helpful (for interpretation) maximum of 100 %. Exact agreement is represented by a $\sigma_{nsb}$ of 0 %.

For a range of Eq. 17 nucleation parameters, Table 1 presents the observation-simulation residuals according to Eq. 18. The temporal profiles of the number size distributions for the entire experiment are shown in Fig. 11, with the model result here from the simulation with minimal residual. In Fig. 11 simulation results are represented by the filled contours, whilst observations are given by the contour lines.

With the shape, size, composition and growth mechanism of the clusters that act as the nucleus of particles subject to ongoing research, in PyCHAM default properties are currently assigned, with a view to advance representation as understanding develops and an appreciation of their physical limitation. An arbitrary involatile component is assumed to form spherical nucleating clusters with a radius of 2 nm. Growth of clusters is assumed to follow gas-particle partitioning (as for particulates of all sizes in PyCHAM). At the current stage of development, this representation of new particle formation in PyCHAM aims to enable simulations of coupled photochemistry and aerosol microphysics in seeded and unseeded experiments. However, a

**Table 1.** Observation-model residuals as defined by Eq. 18 for a dark limonene oxidation experiment without seed particles, with experiment setup described in the main text. Residuals are given for a variety of nucleation parameters, with the minimum residual representing the best fit of simulation to observations.

| $nuc_{v1}$, $nuc_{v2}$, $nuc_{v3}$ | full-moving $\sigma_{nsb}$ (%) | moving-centre $\sigma_{nsb}$ (%) | notes |
|---|---|---|---|
| $2x10^4, -4, 5x10^2$ | 74 | 74 | nucleation duration too long → reduce $nuc_{v3}$ |
| $2x10^4, -4x10^2, 1x10^2$ | 63 | 59 | nucleation commences too late → reduce $nuc_{v2}$ |
| $2x10^4, -4, 1x10^2$ | 53 | 48 | too few particles newly nucleated → increase $nuc_{v1}$ |
| $3x10^4, -4, 1x10^2$ | 40 | 34 | best fit |

**Referee #1, comment #7**

[revised manuscript text omitted]